# A novel proneural function of Asense is integrated with the sequential actions of Delta-Notch, L'sc and Su(H) to promote the neuroepithelial to neuroblast transition

Mercedes Martin, Francisco Gutierrez-Avino, Mirja N. Shaikh¤, Francisco J. Tejedor ᴵᴰ*

Instituto de Neurociencias, Consejo Superior de Investigaciones Científicas and Universidad Miguel Hernandez, Sant Joan d'Alacant, Spain

¤ Current address: Dept. of Biochemistry and Molecular Biology, Shahjalal University of Science and Technology, Sylhet, Bangladesh
* f.tejedor@umh.es

**Data Availability Statement:** All relevant data are within the paper (Figures, Table and text) and the Supporting Information files. In addition, full original images of all main figure image panels of

## Abstract

In order for neural progenitors (NPs) to generate distinct populations of neurons at the right time and place during CNS development, they must switch from undergoing purely proliferative, self-renewing divisions to neurogenic, asymmetric divisions in a tightly regulated manner. In the developing *Drosophila* optic lobe, neuroepithelial (NE) cells of the outer proliferation center (OPC) are progressively transformed into neurogenic NPs called neuroblasts (NBs) in a medial to lateral proneural wave. The cells undergoing this transition express Lethal of Scute (L'sc), a proneural transcription factor (TF) of the Acheate Scute Complex (AS-C). Here we show that there is also a peak of expression of Asense (Ase), another AS-C TF, in the cells neighboring those with transient L'sc expression. These peak of Ase cells help to identify a new transitional stage as they have lost NE markers and L'sc, they receive a strong Notch signal and barely exhibit NB markers. This expression of Ase is necessary and sufficient to promote the NE to NB transition in a more robust and rapid manner than that of *l'sc* gain of function or *Notch* loss of function. Thus, to our knowledge, these data provide the first direct evidence of a proneural role for Ase in CNS neurogenesis. Strikingly, we found that strong Delta-Notch signaling at the lateral border of the NE triggers *l'sc* expression, which in turn induces *ase* expression in the adjacent cells through the activation of Delta-Notch signaling. These results reveal two novel non-conventional actions of Notch signaling in driving the expression of proneural factors, in contrast to the repression that Notch signaling exerts on them during classical lateral inhibition. Finally, Suppressor of Hairless (Su(H)), which seems to be upregulated late in the transitioning cells and in NBs, represses *l'sc* and *ase*, ensuring their expression is transient. Thus, our data identify a key proneural role of Ase that is integrated with the sequential activities of Delta-Notch signaling, L'sc, and Su(H), driving the progressive transformation of NE cells into NBs.

the paper can be downloaded from the repository DIGITAL.CSIC (https://digital.csic.es/handle/10261/336403?mode=full https://doi.org/10.20350/digitalCSIC/15608). Although not necessary to replicate the findings of the paper, any piece of data of the entire raw data set of this work will be made freely available to any researcher interested on them upon request (contact: f.tejedor@umh.es and mercedes.martinf@umh.es).

**Funding:** This work was supported by grants from the Ministerio de Economía, Industria y Competitividad (AEI/FEDER, UE: BFU2016–80273-R) and CSIC (2020AEP180) to FJT. MNS was recipient of a Santiago Grisolia Fellowship from the Generalitat Valenciana. The funders had no role in study design, data collection and analysis, decision to publish, or preparation of the manuscript.

**Competing interests:** The authors have declared that no competing interests exist.

## Author summary

During brain development, neural progenitors (NPs) that initially divide symmetrically to increase in number, gradually switch to a neurogenic (asymmetrically dividing) state. This transition is crucial to generate proper neuronal populations at the right place and time. We have studied how this transition is regulated using the larval optic lobe of *Drosophila melanogaster* (the fruit fly). In this experimental model, neuroepithelial cells (NE) are progressively transformed into neurogenic NPs called neuroblasts (NBs) following a temporal wave that sweeps across the tissue. We have found that Asense, a proneural transcription factor, has a peak of expression following this wave and promotes the NE-NB transition. Our data help to reformulate the working model of this key transition by showing how this novel action of Asense can be non-conventionally integrated into a regulatory network with well-known signaling mechanisms, which may have interesting evolutionary implications. Thus, we propose that the way in which the master regulators of neurogenesis, proneural factors and Notch signaling, interact for the generation of neurogenic NPs is very different in simple nervous systems, where these NPs are individually selected, compared to complex systems in which distinct populations of those progenitors are progressively generated following a neurogenic wave.

## Introduction

The correct formation of a functional nervous system depends on the dynamic coordination of NPs proliferation and differentiation. During the development of a complex central nervous system (CNS), NPs progress sequentially through distinct stages, initially dividing symmetrically to expand their population and later, dividing asymmetrically to produce different populations of neurons in a stepwise manner [1–4]. This transition from proliferating to neurogenic NPs is a key developmental step that is tightly regulated [3,5,6]. Hence, it is important to reveal the full set of genes and molecular mechanisms that control this process.

The larval optic lobe (OL) primordia of *Drosophila melanogaster (Dm)* is an experimental model very well suited to analyze the genetic and molecular basis of this transition, as the spatio-temporal organization of these cellular processes has been clearly defined [4,7–16]. The *Dm* OL originates from an epithelial vesicle that invaginates from the head epidermis and becomes attached to the brain [17,18]. This small cluster of NE cells increases in number during 1st and 2nd instar larvae, and it becomes segregated into two primordia by the end of this period: the Outer (OPC) and Inner (IPC) Proliferation Centers [19,20]. Of these, the OPC gives rise to the precursor cells of the lamina and the outer medulla, while the IPC generates those of the lobula complex and the inner medulla [14,21,22].

The OPC initially grows through symmetric divisions of NE cells. Clonal analysis has shown that NBs originate from these NEs [23]. Thus, during the 3rd larval instar, the OPC NE cells progressively differentiate into medulla NBs and lamina precursor cells (LPCs) at the medial and lateral edges of this structure, respectively. At the same time, NBs switch from symmetric to asymmetric divisions and concomitantly change the orientation of these cell divisions from tangential to radial [8]. The medulla NBs divide asymmetrically in a self-renewing fashion producing a new NB and a ganglion mother cell (GMC), the latter dividing once to generate two medulla neuronal precursors, formerly called ganglion cells (GCs) [8]. These sequential events are reminiscent of those taking place in the developing mammalian cerebral cortex and neural tube [7,24–26]. In these mammalian tissues, the NE cells in the ventricular zone (VZ) initially divide symmetrically to expand the pool of progenitors and then, at the

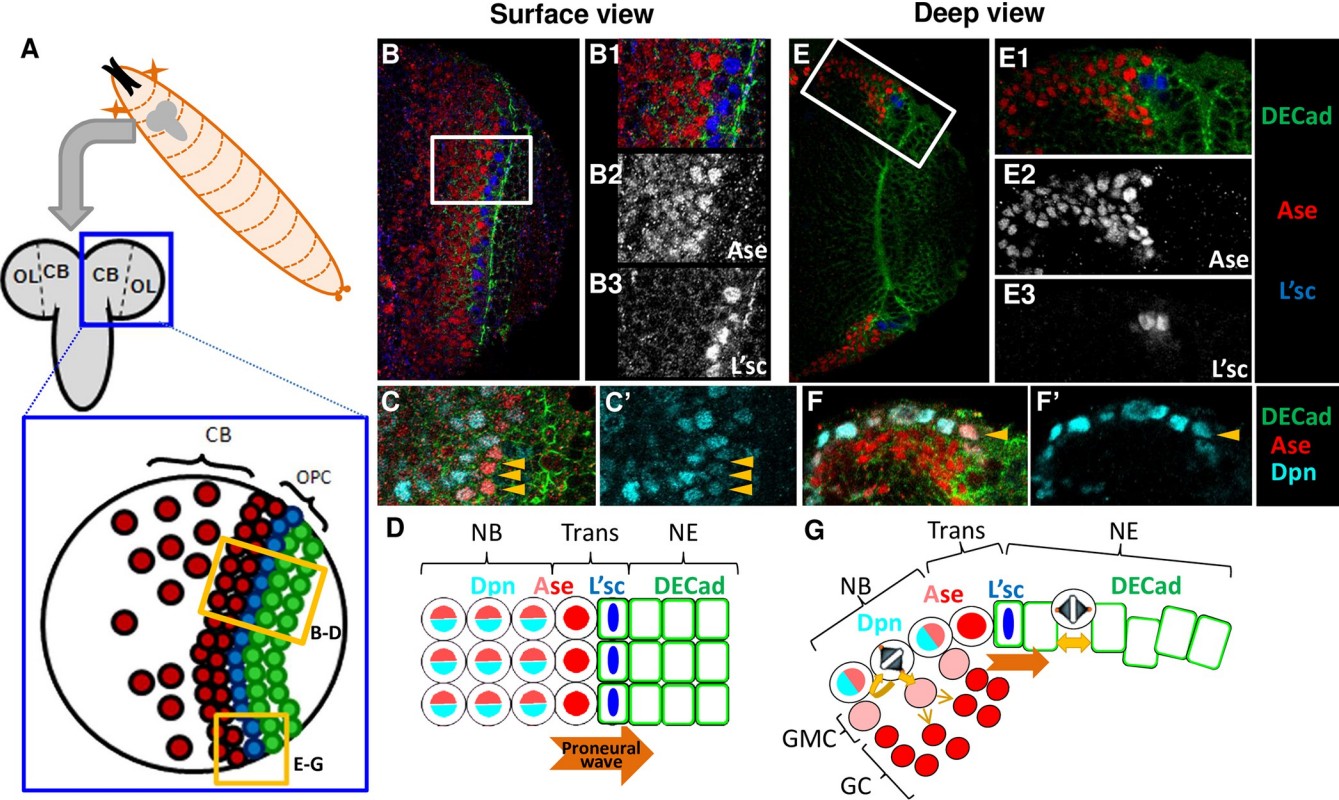

**Fig 1. Expression of Ase in the OPC. A**. Scheme representing the ventral view of the larval *Dm* brain indicating the location of the CB, OL and OPC regions, as well as the position of the NE cells (green), NE-NB transition cells (blue) and NBs (red) within the OPC. **B, C, E, F**. Confocal images showing the expression of Ase, DECad, Dpn and L'sc in surface and deep layers, and high magnification views of the boxed areas, as indicated. Note that the strongest Ase immunostained cells are neighbors of the L'sc[+] cells and they have little Dpn (arrowheads). **D, G**. Schemes summarizing the data depicted in the preceding panels.

onset of neurogenesis, they progressively lose their NE properties and are transformed into radial glial cells that most often divide asymmetrically to generate a new radial glial cell and an intermediate progenitor that move into the subventricular zone (SVZ) where it divides terminally into two neurons [3].

The NE to NB transition in the medulla OPC progresses in a spatially and temporally ordered manner, following a medial to lateral direction [9] (see Fig 1A, 1D and 1G for schematic representations). This stepwise transition is crucial to progressively generate the different populations of medulla neurons [27,28] and it requires the coordinated action of several signaling pathways, mainly JAK/STAT, EGFR, Fat-Hippo and Notch [10,15,28]. Thus, JAK/STAT signaling is activated in the lateral NE cells of the OPC, initially promoting NE expansion and repressing the progression of the NE-NB transition [29–31]. EGFR signaling is required for the proliferation of NE cells and to induce the expression of the L'sc proneural factor [32] that precedes NB formation [31]. L'sc promotes cell-autonomously the expression of Delta (Dl), which in turn activates the Notch pathway in adjacent cells on both the medial (NB) and lateral (NE) sides. While the down-regulation of Notch signaling seems to be required for the transition to NBs, the activation of Notch in NE cells represses *l'sc* expression and prevents a premature switch to NBs [29,32–36]. Hence, the coordination of these signals produces a wave of *l'sc* expression that sweeps across the NE in a medial to lateral direction. This is named the proneural wave [31,32,37].

Although *l'sc* expression seems to be sufficient to induce the NB fate [31,33], its loss of function (LoF) does not prevent the NE-NB transition but rather, it simply alters the timing of NB differentiation (31). Moreover, it remains unclear how L'sc induces NB fate. L'sc belongs to the AS-C family of basic helix-loop-helix (bHLH) transcription factors (TFs), the products of the *achaete (ac)*, *scute (sc)*, *l'sc*, and *asense (ase)* genes [38–41]. Classic studies of these genes showed they are required for the commitment of epithelial cells to the NP lineage in the *Dm* peripheral nervous system (PNS) and CNS [42–45].

There is some redundancy in the roles of the AS-C TFs in the NE-NB transition [31]. In addition to L'sc, other AS-C proteins are expressed in the OL, including Sc in NE cells and NBs [9], and Ase in NBs and their GC progeny [31,46–48]. Remarkably, *ase*-deficient flies develop a defective adult OL [47], although NB formation did not seem to be overtly affected [31]. Nevertheless, the delay in NB formation associated with the loss of *l'sc* appears to be enhanced by the deletion of *ase* [31].

Ase plays an important role in regulating proliferation in the OL [49], particularly, the cell cycle exit and terminal differentiation of new born neurons [48]. However, its role in NB formation remains unclear and no proneural role for this gene has so far been reported in the CNS. Notably, DamID analysis and expression profiling of *ase* mutant embryos predict a dual role for Ase. On the one hand, Ase could activate the expression of self-renewal genes and repress the expression of differentiation genes in NBs, while on the other hand, it could promote the neuronal differentiation of the NB progeny [50]. Interestingly, Ase plays a proneural role in the generation of bristles at the anterior wing margin [51]. Moreover, Ascl1/Mash1, its closest vertebrate orthologue, plays an important proneural role in CNS neurogenesis [52–55].

In the light of the above, we set out to assess a possible role of Ase in the NE to NB transition. Accordingly, we studied the expression of Ase during this transition in detail, and the time course of the alterations caused by its gain-of-function (GoF) and LoF relative to the equivalent genetic conditions for *l'sc*, as well as for Notch downregulation. Furthermore, we analyzed how the expression of Ase and L'sc is regulated, and how Ase activity integrates with L'sc and Notch signaling in the context of the NE to NB transition.

## Materials and methods

### *Drosophila melanogaster* strains

Fly stocks were raised at 25°C on standard medium. The following strains were used: *Oregon R* (wild type -wt); *ase¹* (formerly known as *sc²*, [47]); *E(spl)mγ-GFP* [56]; *hs FLP; act-FRT-y +-FRT Gal4 UAS GFP* (BDSC stock # 30558); *Su(H)-LacZ* (reporter 1, *P{E(spl)m8-HLH-lacZ. Gbe}3*; expresses beta-galactosidase under the control of Su(H) binding sites from E(spl) m8-HLH and Gbe [57]); *Su(H)-LacZ* (reporter 2; BDSC stock # 10689; [58]); *UAS-ase* [59]; *UAS-ase-RNAi* (line 1:VDRC stock # GD12444; [60]); *UAS-ase-RNAi* (line 2: BDSC stock # 44552; [61]); *UAS-Dl* (BDSC stock # 26694); *UAS Dl-DN* (BDSC stock # 5613, [62]); *UAS-dGFP* [63]; *UAS-Dpn-RNAi* (BDSC stock # 26320; [61]), *UAS-L'sc* [64]; *UAS-L'sc-RNAi* (VDRC stock # v104691); *UAS-Su(H).VP16* (BDSC stock # 83149; [65]), *UAS-Su(H)-RNAi* (BDSC stock # 28900; [61]).

### Gene misexpression and RNAi using the UAS/Gal4 system

The *ase-Gal4* [66], *c855a-Gal4* [67], and *c820-Gal4* [68] lines were used to induce ectopic protein expression or RNAi. The patterns of expression of *c855a-Gal4* and *c820-Gal4* at the NE-NB transition are depicted in S1 Fig. Larvae carrying Gal4/UAS constructs were kept at a restrictive temperature (16–17°C), at which no consistent transgene induction or concomitant phenotypes were detected (S2A–S2D Fig). At the appropriate stage of development, the

temperature was increased to 29–30˚C until larvae reached the wandering stage, at which point they were dissected for analysis. Ase, Su(H) and L'sc RNAi induction was maintained for 24-36h, while *Notch RNAi, ase, Dl-DN*, and *l'sc* misexpressions were induced for 8-12h in the late 3rd instar, or for 36h around the beginning of 3rd instar, as indicated in each experiment.

## Generation of clones

Recombinant clones were generated using the Flip-out technique [69]. To this end, *hsp70-Flp; Actin5C <yellow⁺, [stop]> Gal4, UAS-GFP* female flies were crossed with males carrying the different UAS constructs indicated in each experiment. Larvae were raised at 25˚C and clones were generated by inducing FRT-mediated recombination with a 10 min heat shock at 37˚C. Subsequently, the larvae were incubated at 29˚C for 16h before dissection.

## Immunohistochemistry (IHC)

Third instar larval brains were dissected out in PBS and fixed in 4% paraformaldehyde (PFA) for 20 min at room temperature (RT), or for 1h at 4˚C (for Dl immunostaining). The brains were then washed with PBST-0.5% (0.5% Triton X-100 in PBS) and non-specific binding was blocked at RT with 10% normal goat serum (NGS) in PBST-0.5% + 0.02% sodium azide for 1h (or 4h for Dl immunostaining). The brains were probed with the primary antibodies overnight at 4˚C and for 1h at RT the next day. After washing in 0.5% bovine serum albumin (BSA) in PBST-0.2%, the brains were then incubated for 1-2h at RT with fluorescent conjugated secondary antibodies (Alexa-488, Alexa-594, Alexa-647, Cy3 or Cy5: Jackson ImmunoResearch) diluted 1:500. Samples were then washed sequentially at RT with BSA/PBST-0.2%, PBST-0.3% and PBS for 30 min each, and finally mounted in Vectashield H-1000 medium (Vector Lab, Germany). Control and mutant samples were processed in parallel under the same conditions.

The primary antibodies used were: rabbit anti-Ase (1:400, a generous gift of A. Carmena [70]); rabbit anti-Ase (1:2000, a generous gift of Y.N. Jan: [46]); rat anti-DECad (1:120, DSHB, Clone DCAD2 [71]); mouse anti-Dl (1:30, DSHB, Clone C594.9B: [72]); guinea pig anti-Dpn (1:3000, [70]); rabbit anti-βGal (1:800, Cappel); chick anti-GFP (1:3000: AvesLab); rabbit anti-GFP (1:800; A1112, Invitrogen); guinea pig anti-L'sc (1:1500, a generous gift of M. Sato [73]); mouse anti-Mira (1:100, a generous gift of F. Matsuzaki, Clone PLF8); rabbit anti-PatJ (1:200; a generous gift of H. Bellen; [74]); mouse anti-Pros (1:30; DSHB, clone MR1A: [75]).

## Image processing, quantification and statistics

Immunostaining was examined by confocal microscopy (Olympus FV10i fluoview or Zeiss LSM 880-Airyscan Elyra PS.1.), analyzing the images with FV10-ASW 4.2 viewer, ZEN lite 3.2 blue edition, or with ImageJ 1.52n. All the samples in each experiment were analyzed in the same session, using the same confocal microscopy acquisition parameters. Equal processing was applied to control and mutant images that were selected from equivalent positions in the OL comprising equal digital area size.

To assess the changes in Ase protein expression, ImageJ 1.52n software was used to measure the mean fluorescent intensity of the nuclei in confocal images. The intensity of the peak of Ase cells was compared with that of the neighboring NBs or the average of medulla NBs, while the intensity of NE cells was taken as the background signal. The same workflow was used to measure L'sc misexpression in *c855 Gal4>UAS l'sc* tissue, comparing the intensity of L'sc immunostaining in NE cells (ectopic) with that of transition cells (endogenous), while the intensity in NBs was taken as the background signal. Data were analyzed in Microsoft Excel 2010.

The number of labelled cells was quantified manually in serial confocal sections taken every 2 μm in the 20–60 μm Z-axis from the ventral surface of the OL using the FV10-ASW 4.2 viewer software. The data was analyzed using Microsoft Excel 2010 and SigmaStat, employing Student t-tests for samples drawn from normally distributed populations with the same variances or alternatively, Mann-Whitney Rank Sum tests for non-normal populations or populations with unequal variances. Differences in both cases were considered significant at $P < 0.05$.

## Results

### *ase* expression peaks at the NE-NB transition

Ase has commonly been used as a molecular marker for central brain (CB) type I and OPC NBs in the larval brain [9,31,76]. Nevertheless, we previously detected a transient *ase* expression in newborn GCs, where it is expressed more strongly than in the dividing NBs [48,77]. To precisely define the expression of *ase* during the NE-NB transition, we analyzed Ase immunostaining in *wt* brains relative to different markers of NE cells (DE-Cadherin, DECad), transition cells (L'sc) and NBs (Deadpan, Dpn; Miranda, Mira: Figs 1B–1G, 2E and 2G). As described previously [46,48,77], Ase was found in all NBs but not in NE or transition (L'sc[+]) cells. Nevertheless, we noticed that the cells neighboring the L'sc[+] transition cells were more strongly immunostained for Ase than the NBs (an 85% increase in intensity; 6/6 brains: Fig 1B–1E). Moreover, these cells expressed NB markers weakly or not at all (Figs 1C,1F; 2E and 2G), suggesting that these cells might still be in a transition phase (see Fig 1D and 1G for a schematic summary). Henceforth, we will refer to these high-level Ase-expressing cells (i.e. strong initial upregulation of *ase*) as the "peak of Ase" cells.

### Ase is necessary for proper NE-NB transition

We wondered if this oscillation of *ase* expression might be implicated in the NE-NB transition. To investigate this possibility, we analyzed the expression of differential markers along the transition under conditions of *ase* GoF and LoF. We first studied *ase[1]*, a deletion of the gene that does not affect any other component of the AS-C [47]. We found that Mira (NB marker) and DECad (NE marker) were co-expressed in many cells during the NE-NB transition (Fig 2A,2B and 2I; S1 Data), and that many L'sc[+] cells were located more towards the middle of the NE rather than at its edge, as it happens in *wt* brains (Fig 2A and 2B; 9/9 brains). Additionally, we observed a mismatch in the apical expression of the NE markers DECad and PatJ, which are normally co-expressed in *wt* brains (Fig 2C and 2D; 7/7 brains). Hence, Ase appears to be required for the correct NE-NB transition.

We next assessed whether the peak of Ase expression is required for the correct NE-NB transition by driving a*se-RNAi* in the most medial NE and transition zone cells using the *c820-Gal4* line (S1A1, S1A2 Fig) with two independent *UAS-ase-RNAi* lines. In these animals we observed almost the absence of peak of Ase cells close to the NE. Furthermore, all the Ase[+] cells at the surface of the OPC had weak Ase labeling and co-expressed Mira (Fig 2E–2H; S3A–S3C Fig; S1 Data). Remarkably, these *ase-RNAi* conditions caused the same phenotype as the *ase[1]* mutation, with a significant increase in the number of cells co-expressing NE and NB markers (Fig 2E–2I; S3A, S3B, S3D Fig; S1 Data). Hence, we concluded that the peak of Ase expression is crucial for the correct NE-NB transition.

### Ase promotes the NE-NB transition

To further analyze the implication of Ase in the NE-NB transition, we misexpressed *ase* in the OPC and IPC NE cells using the *c855a-Gal4* driver ([67]; S1B Fig). First, we induced *ase* expression from the beginning of the 3rd instar, a period when the NE-NB transition has barely

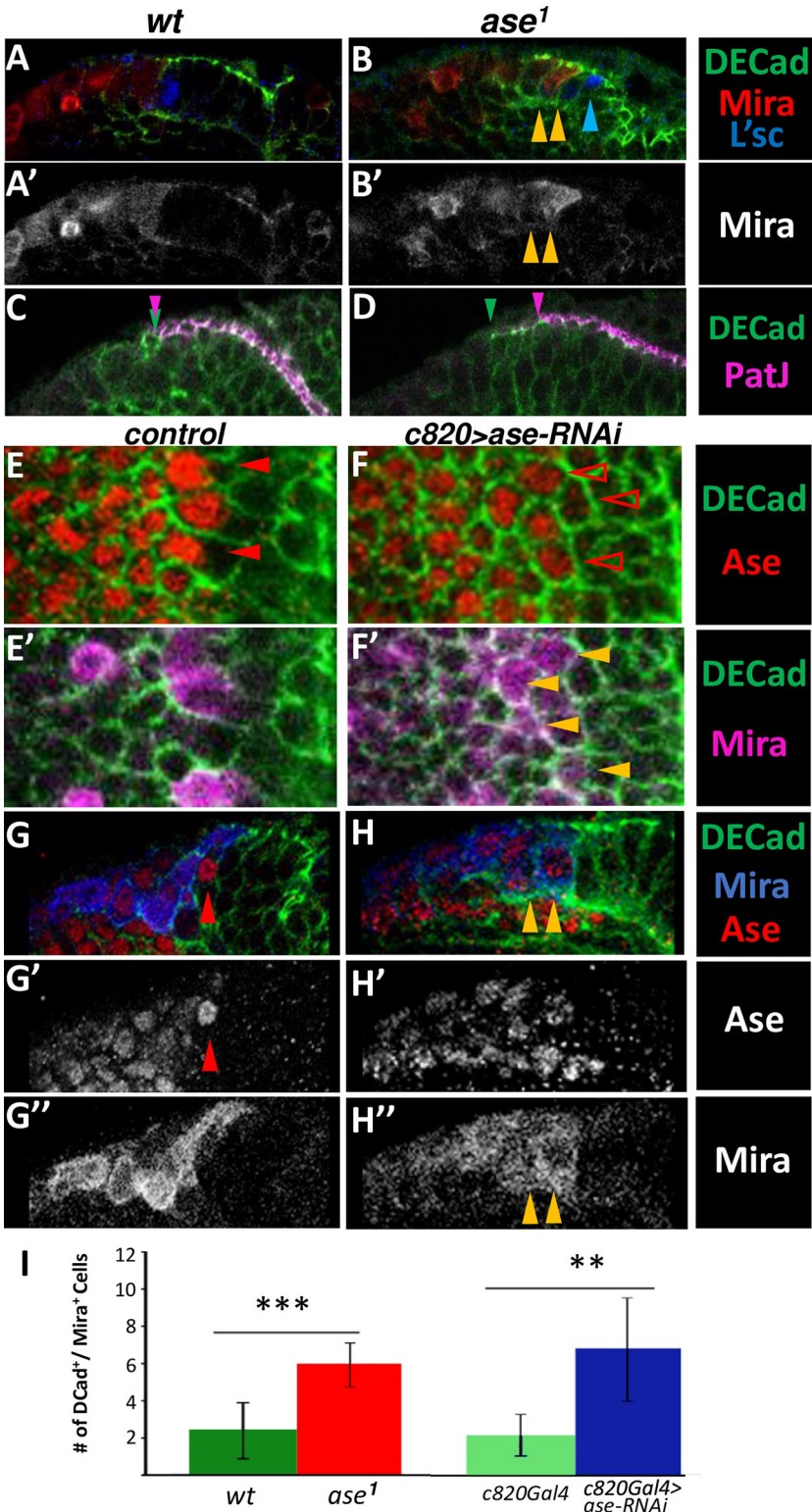

**Fig 2. Alterations to the NE-NB transition by *ase LoF*. A-F**: High magnification confocal images of deep OPC layers of *wt*, *ase[1]*, *c820 Gal4 (control)* and *c820 Gal4; UAS ase-RNAi* flies showing immunostaining for Ase, DECad, Mira, L'sc and PatJ as indicated. **A, B**. Note that in the *ase[1]* sample there are cells co-expressing DECad and Mira (yellow arrowheads) whereas they are segregated in the control. In addition, the *ase[1]* sample has L'sc stained cells in the middle of the NE (blue arrowhead) instead of at its medial edge, as in the *wt* larvae. **C, D**. Neuroepithelial cells of *ase[1]* present a

mismatch in the DEcad and PatJ cell expression patterns, compared to the control sample as depicted by the green and magenta arrowheads in. **E-H**. Surface (E,F) and deep (G,H) views. The *c820 >ase-RNAi* sample lack peak of *ase* cells compared to the control (red arrowhead) and exhibits weakly labeled Ase+ cells co-expressing Mira and DECad (yellow arrowheads). Also note that the peak of *ase* cell in the control practically lacks Mira labeling (G, red arrowhead). **I**. Quantification of Mira/DECad co-expressing cells in *wt (n = 12)*, *ase¹ (n = 16)*, *c820Gal4 (n = 5)* and *c820 > ase-RNAi* (n = 5) OLs. Statistical significance was assessed with the Mann Whitney test: *wt vs. ase¹* (P<0.001) and *c820Gal4 vs. c820>ase-RNAi* (P = 0.012).

commenced (see Materials and Methods). This misexpression induced a decrease in the size of the OL and remarkably, the almost complete elimination of OPC NE cells at the expense of NBs, while neuronal precursors (Pros⁺ GCs) were situated normally within the OPC (Fig 3A and 3B; Table 1). These data strongly suggest that the misexpression of *ase* in NE cells induces a premature generation of NBs, thereby reducing the pool of NE cells and, consequently, the size of the OL.

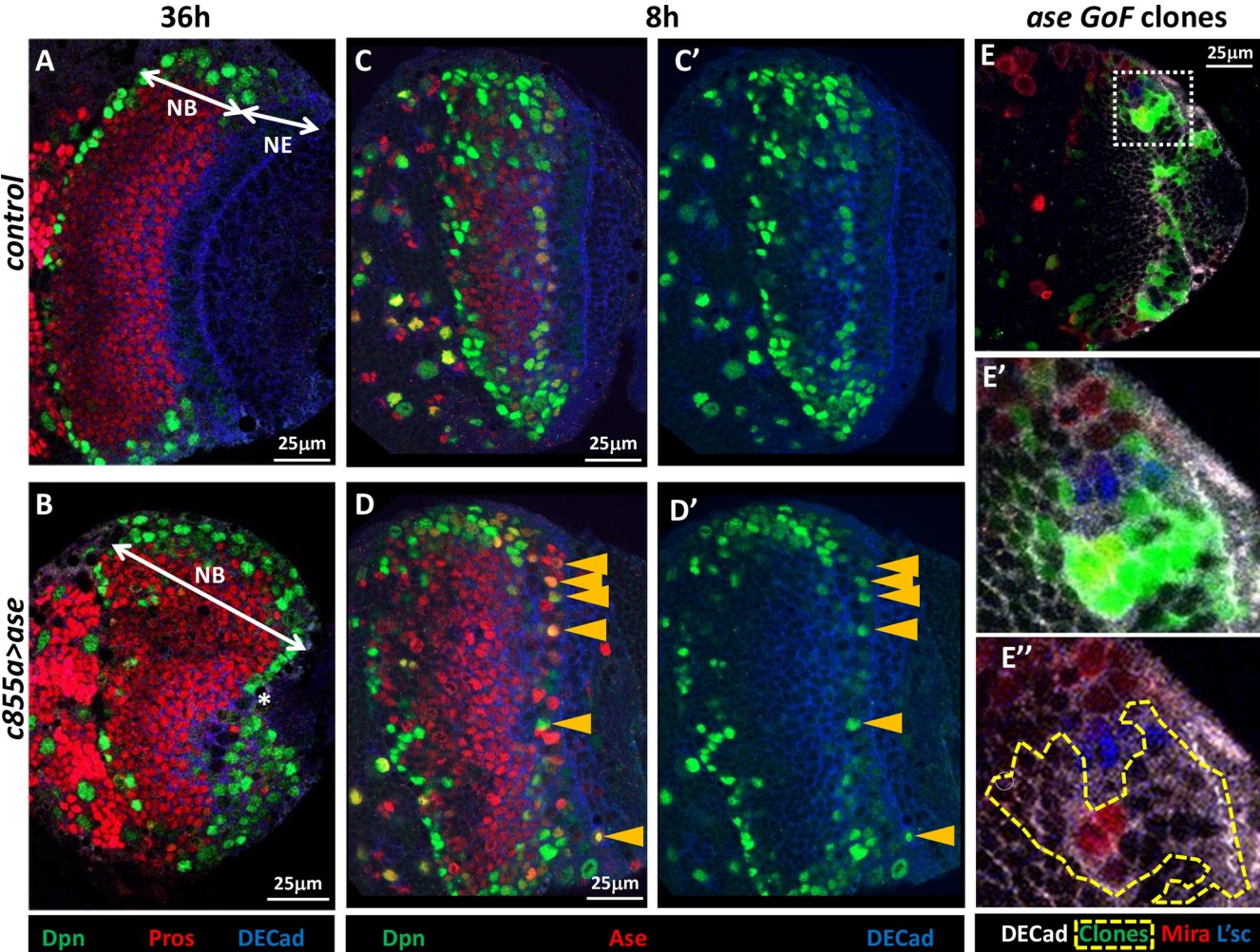

**Fig 3. Misexpression of Ase in the NE generates NBs. A, B**. Confocal images close to the surface of the OPC of control (*c855a Gal4*) and *c855a Gal4/UAS ase* larval brains after a 36h induction. Note that the OL size is reduced in the *c855a> ase* specimen relative to the control and that there are no NE (DECad⁺) cells. Moreover, the most lateral region (NE in the control sample) is occupied by NBs (Dpn⁺) in the *c855a>ase* sample while in both specimens the interior area of the OPC is full of GCs (Pros⁺). **C, D**. Similar experiment after an 8h induction. Note the numerous intermingled Dpn⁺/Ase⁺ cells (yellow arrowheads) in the NE of the *c855a>ase* sample relative to the control. **E**. Ase misexpressing (16h) clones originated in the NE. **E', E"**. Magnification of the white framed area in **E**. Notice that the clon shows altered DECad labeling and contains several Mira⁺ cells inside but there are no ectopic Mira+ cells at its outside border, and it does not exhibit L'sc immunostaining.

To assess if Ase could directly promote the transformation of NE cells into NBs, *Ase* was misexpressed in the NE for a short period of time (8h) at the end of 3rd larval instar using the same *c855a-Gal4* line. Interestingly, this misexpression produced ectopic cells expressing NB markers (Dpn and Mira) within the NE and at its edge (Fig 3C and 3D; S3A Fig; S1 Data; Table 1). These observations led us to assess whether Ase promotes NB formation in a cell autonomous or non-cell autonomous manner. Thus, we generated small clones of cells expressing high levels of Ase and we observed ectopic expression of the NB marker Mira in 59% of the clones that were situated within the NE while none of these clones exhibited ectopic Mira at their edge (Fig 3E–3E"). Furthermore, no increase in L'sc expression was detected when Ase was misexpressed in the NE (Fig 3E–3E"; S4A–S4D Fig). Together, these results demonstrate that Ase is sufficient to transform NE cells into NBs possibly in a cell autonomous manner and suggest that this action is not mediated by L'sc.

## L'sc is not sufficient to promote NE-NB transition at the OPC

The proneural factor L'sc was reported to be necessary for the timely onset of NB differentiation and furthermore, its ectopic expression was sufficient to induce the appearance of NBs. Accordingly, it was proposed that the transient expression of L'sc signals the transition of NE cells to NBs in the OPC [31]. Strikingly, we found that L'sc is still detected transiently in individual cells in *ase¹* brains (Fig 2B) despite the strong alterations to the NE-NB transition observed (Fig 2A–2D). Moreover, Ase misexpression appears to strongly induce the NE-NB transition without affecting L'sc expression (Fig 3E; S4A–S4D Fig). Therefore, we decided to assess the capacity of L'sc misexpression to induce NB formation in the OPC under the same conditions as those employed to study Ase. Thus, when c855a-Gal4 was used to drive strong L'sc expression for 36h, very few ectopic cells expressing NB markers were observed within the NE. Moreover, no obvious changes in the structure or size of the NE or OL were detected relative to control brains (Fig 4A and 4B; Table 1). Furthermore, a short (8h) induction of L'sc in the NE, which reached similar (or higher) levels as the endogenous expression in transition cells (S2E Fig; S1 Data), did not trigger NB marker expression in the medulla NE (Fig 4A and 4C; S4E-S4H Fig; Table 1) although we detected induction of Mira in the Lamina NE (S4E–S4H Fig). Therefore, we concluded that Ase misexpression in the medulla NE has a more rapid and robust proneural effect than that of L'sc.

## The down-regulation of Notch induces the NE-NB transition at a slower rate than the gain of function of *ase*

It has been proposed that the NE-NB transition is triggered by the inhibition of Notch activity in NE cells induced by L'sc or through the negative feedback loop with Delta [29,33,34,76]. In order

**Table 1. Time dependent phenotypes of Ase, Dl-DN, L'sc and N RNAi missexpression in the OPC NE.**

| Time of missexpression | Genotype | NB marker⁺ cells in NE | | Reduced NE | | Smaller OL | | Figure Data |
|---|---|---|---|---|---|---|---|---|
| 8h | c855a>Ase | **** | 10/10 | | 0/18 | | 0/18 | Fig 3; S4 Fig |
| | c855a>L'sc | * | 1/6 | | 0/6 | | 0/6 | Fig 4; S4 Fig |
| | c855a>N RNAi | * | 1/14 | | 0/14 | | 0/14 | S6 Fig |
| | c855a>DL-DN | * | 4/15 | * | 2/21 | * | 1/21 | Fig 4 |
| 36h | c855a>Ase | **** | 10/10 | **** | 10/10 | **** | 10/10 | Fig 3; S4 Fig |
| | c855a>L'sc | * | 8/8 | * | 4/8 | | 0/8 | Fig 4 |
| | c855a>N RNAi | ** | 14/14 | *** | 14/14 | | 0/14 | S6 Fig |
| | c855a>DL-DN | **** | 19/19 | **** | 18/19 | * | 2/19 | Fig 4 |

Comparative summary of the results presented in the corresponding figures (right column) as described in the main text. Expressivity (in an arbitrary scale: *, subtle, **, substantial, ***, strong, ****, very strong) and penetrance (ratio of analyzed brains) are shown in parallel columns for each phenotype/genotype generated by driving *UAS-Ase*, *UAS-Dl-DN*, *UAS-L'sc* and *UAS-N RNAi* with the *c855a-Gal4* driver during two time periods (8 and 36 h) in late third instar larvae

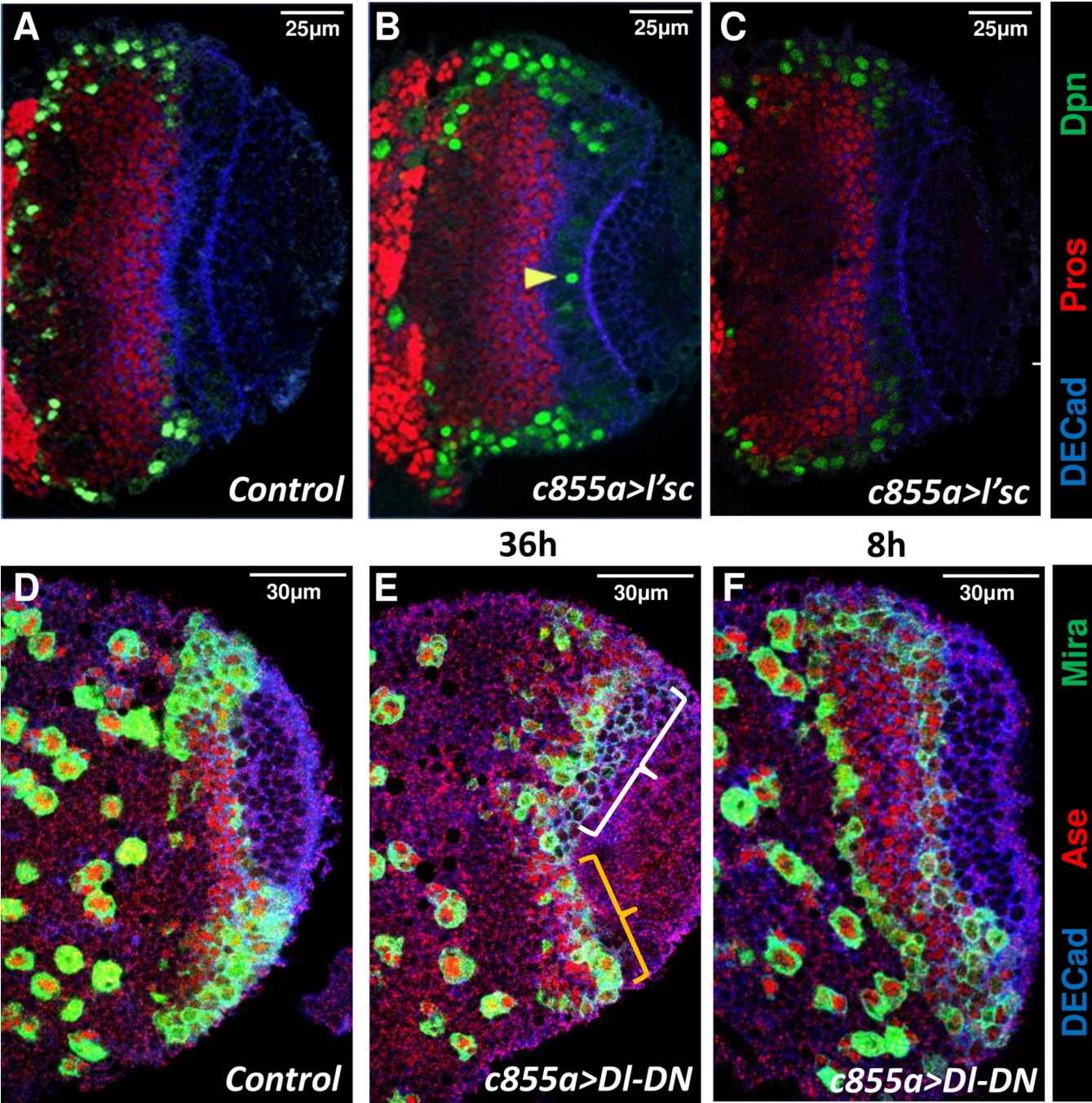

**Fig 4. L'sc misexpression and Notch down regulation in the NE generates NBs at low rate.** Confocal images acquired close to the surface of the OPC of control (*c855a Gal4*) (**A, D**), *c855a Gal4/UAS l'sc* (**B, C**) and *c855a Gal4/UAS Dl-DN* (**E, F**) larval brains after a 36h (**B, E**) or an 8h (**C, F**) induction. **B**. Note that compared with *ase* misexpression samples (Fig 3A and 3B) the NE has a relatively normal shape and contains very few Dpn+ cells (yellow arrowheads). **C.** There are no Dpn+ cells in the NE after *l'sc* misexpression (*c855 > l'sc*) for 8h compared to the large number observed after *ase* misexpression in the same time period (compare to Fig 3D). **E.** Long term Notch LoF with Dl-DN produces a loss of NE cells (yellow bracket), but the size of the OL is maintained. Also notice the large number of DECad+ cells that coexpress Mira but no Ase within the white braket area. **F,** Short term Dl-DN induction does not produce phenotype.

to distinguish between inductive and permissive actions, we decided to compare the time course of the phenotypic changes induced by inhibiting Notch in the NE to that of *ase* GoF. We assumed that an inductive factor/signal should produce an immediate effect (i.e. upregulation of NB

markers) in NE cells. In contrast, a permissive factor would prime the cells until they are reached by the inductive signal at the proper position of the neurogenic wave and, consequently, these concomitant actions should require longer time to produce the effect. To emulate the down-regulation of Notch protein that takes place at the proneural wave front [78], we performed N RNAi and we expressed a Delta dominant negative (Dl-DN) construct. The truncated Dl protein produced by this construct interacts intracellularly with Notch, preventing it from reaching the surface of the cell and thereby suppressing Notch activity in a cell autonomous manner [62,72,79]. When *c855a-Gal4* was used to drive Dl-DN expression in the NE there was a rapid and strong depletion of the Notch protein (S5A and S5B Fig; 21/21 brains), although we detected only subtle alterations in the NE-NB transition over the short term relative to the *c855a>ase* larvae (Table 1; compare Fig 3C and 3D with Fig 4D and 4F). In contrast, N RNAi induced for a short term with the *c855a-Gal4* driver did not cause any detectable downregulation of Notch or alterations in the NE-NB transition (S6A and S6B Fig; Table 1). Nevertheless, the depletion of Notch in the NE throughout the neurogenic period in the OPC (36h) using either Dl-DN or N RNAi caused extensive co-expression of NE and NB markers, and a concomitant reduction in the NE area (Figs 4D and 4E; S6A and S6C; Table 1). Intriguingly, despite the obvious reduction in their OPC NE, *c855a>Dl-DN* and *c855a>N RNAi* larvae did not exhibit an overly small OL, in clear contrast to what we observed in *c855a>ase* larvae (Table 1; compare Figs 3A, 3B, 4D, 4E and S6A, S6C Fig). Together these experiments demonstrate that the *GoF of ase* is capable of inducing a more rapid and efficient NE-NB transition than the down-regulation of *Notch*.

Remarkably, in contrast to the normal NE-NB transition, a large proportion of the most lateral NB cells expressed very little or even no Ase in *c855a>Dl-DN* larvae (Fig 4D, 4E; S5C, S5D Fig; 16/16 brains). Furthermore, the induction of Dl-DN in the NE for a short period was sufficient to substantially reduce the number of peak of Ase cells (S5E Fig; S1 Data). These data strongly suggest that the down-regulation of *Notch* represses or to some extent delays the expression of *ase* in the NE-NB transition.

## L'sc promotes *ase* expression in a non-cell autonomous manner during the NE-NB transition

Considering that the transient expression of L'sc precedes the peak of Ase during the NE-NB transition, we wondered if L'sc might be involved in upregulating *ase* expression. To this end, we used RNAi to down-regulate L'sc expression in transitioning cells using the *c820-Gal4* line and we observed a strong suppression of the peak of Ase expression (Fig 5A and 5B; 6/6 brains). When *ase* expression was analyzed in the NE following L'sc misexpression driven by the *c855a-Gal4* line, very few scattered Ase-expressing cells within the NE were found (S5F and S5G Fig). In order to examine whether the activation of Ase by L'sc occurs in a cell- or non-cell autonomous manner, L´sc expression was induced in small clones, paying particular attention to clones located within the NE or at its medial border. Enhanced Ase expression was only observed at the outer border of the clones and always outside the NE (Fig 5C and 5D; 26/32 clones). Furthermore, native *ase* expression was abolished within the clone when it covered the transition zone (Fig 5D'). Together, these results indicate that although L'sc alone cannot promote Ase expression in NE cells, it is required to induce it in transitioning cells in a non-cell autonomous manner.

## Delta-Notch signaling promotes and Su(H) down-regulates Ase expression at the NE-NB transition

Notch signaling is fundamental for the progression of the proneural wave and its activation follows a very dynamic pattern during the NE-NB transition [29,32–36,76,80]. Notch activity

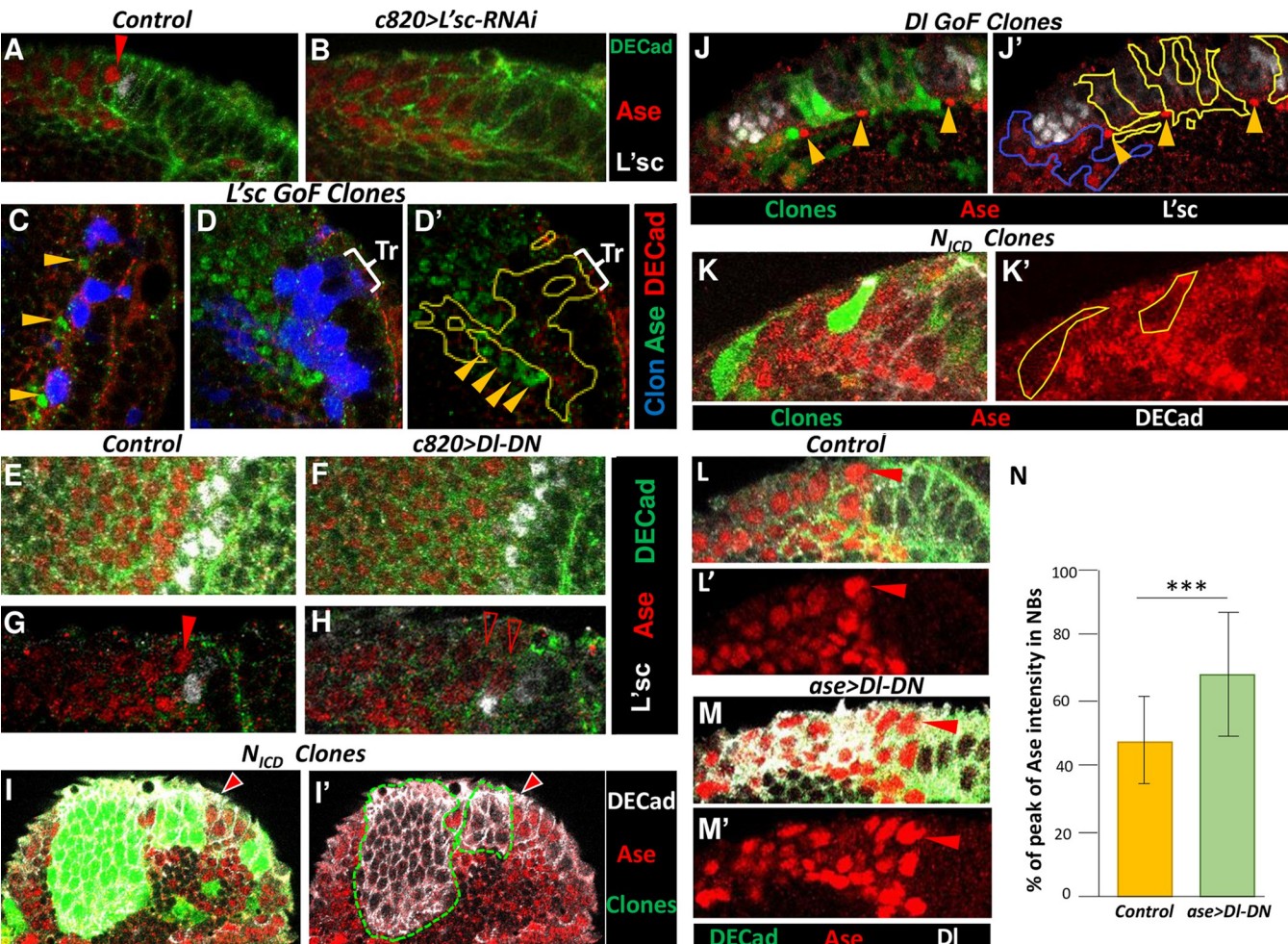

**Fig 5. Effects of L'sc and Notch LoF and GoF on Ase expression.** High magnification views of confocal images of the NE-NB transition **A,B.** Effect of *l'sc* RNAi at the NE-NB transition. Notice the absence of peak of Ase cells (red arrow) in the *c820>L'sc-RNAi* sample compared to the control. **C, D**. Clonal analysis of *l'sc* misexpression. **C**. Three small L'sc GoF clones located at the edge of the NE exhibit strong Ase⁺ cells (arrowheads) located at their outer medial border. **D, D'.** A large clone expanding the NE and the putative transition zone (Tr) does not contain Ase⁺ cells but there are several Ase⁺ cells (arrowheads) located outside, adjacent to its border. **E-H.** Effect of Notch downregulation at the NE-NB transition. Surface (E,F) and deep (G,H) views. The OPC of a *c820>Dl-DN* larvae exhibits a strong decrease of Ase expression in the medial L'sc⁺ neighboring cells (arrowheads) relative to the control. **I, I'.** Effect of *Notch GoF*. Two $N_{icd}$ clones (16h after induction) expanding the medulla OPC NE show expression of DECad and lack Ase labeling. The arrows point to the NE-NB borders. **J, J'.** Effect of Delta-Notch signaling. Several small *Dl GoF* clones located at the OPC NE (yellow line) lack Ase immunostaining but there are strong Ase labelled cells located adjacent to them (yellow arrowheads). In contrast, a clone located in the NB region (blue line) contains many Ase expressing cells. **K, K'.** Two $N_{ICD}$ clones located in NBs are deficient in Ase labeling. **L, M**. Effect of Dl-DN driven in NBs. Deep views of control (*Ase-Gal4*) and *ase>Dl-DN*. Notice the strong increase of Ase labeling in NBs of the *ase>Dl-DN* sample related to its Ase peak cell (arrow) compared to the control. **N**. Quantification of Ase labeling intensity expressed as the % of the peak of Ase intensity in NBs. Differences are significant (t-Student test P<0.001).

is linked to the expression of L'sc, which according to the current model promotes cell-autonomously the expression of *Dl* that in turn activates Notch signaling in the cells adjacent on both sides of the L'sc⁺ transition cell (Fig 6A and 6F). We observed that the majority of the Ase peak cells in the OPC of *wt* larval OLs were neighboring cells with very high expression of Dl at the NE edge (Fig 6B and 6B'; 85% of cells, 9/9 brains). Thus, we wondered whether Delta-Notch signaling regulates *ase* expression during the NE-NB transition. To evaluate this possibility, we studied the correlation between *ase* expression and Notch activation using an E(spl)mγ reporter that drives GFP expression in cells in which the Notch pathway is activated [36,56]. Remarkably, we observed strong Notch activity in the peak of Ase cells (Fig 6C; 77% of cells, 7/

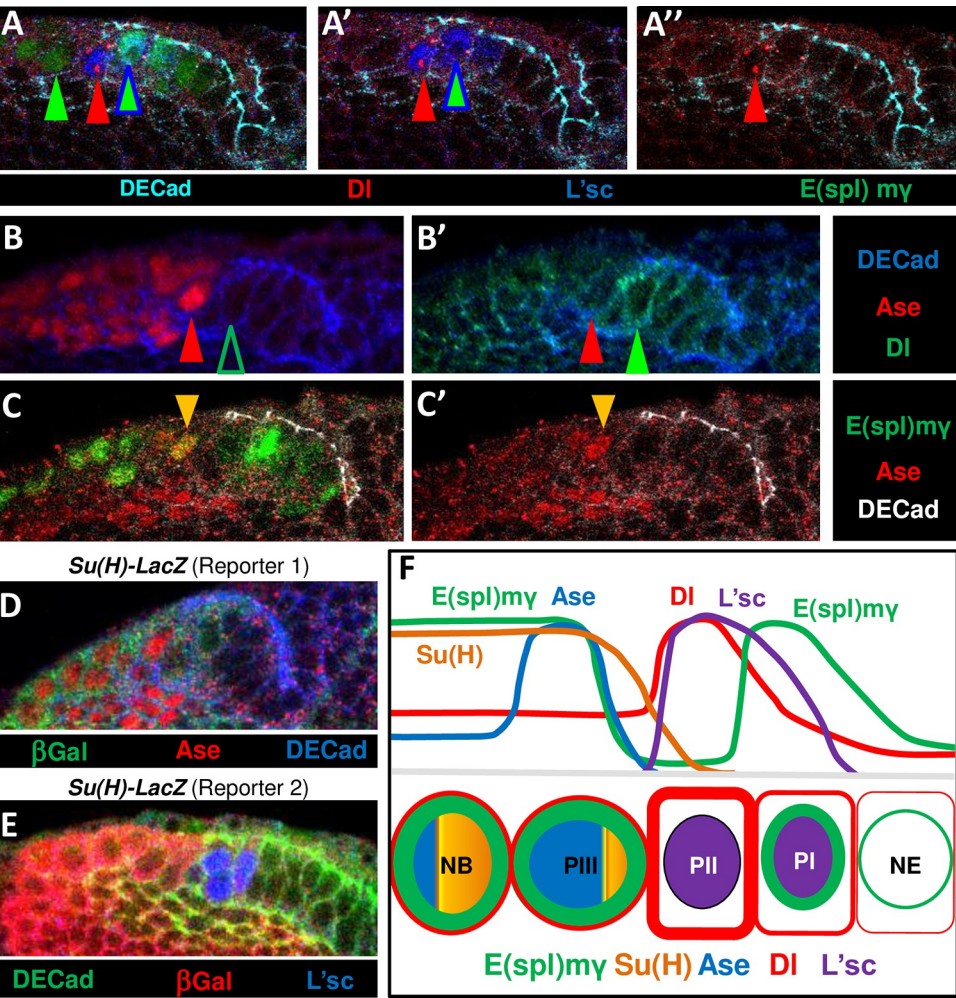

**Fig 6. Pattern of expression of Ase, L'sc and Notch signaling elements at the NE-NB transition. A-C** Represetative high magnification view of a deep confocal image of the OPC of *wt* or *E(spl)mγ-GFP* larval brains **A, A', A".** Images showing the coexpression of L'sc and Dl in the transition cells and the expression of E(spl)mγ-GFP that reveals high Notch activity in the cells (green arrows) neighboring the strong Dl expressing cell (red arrow). Remarkably, the most medial L'sc⁺ transition cell (blue/green arrowhead) expresses E(spl)mγ, revealing a strong Notch signaling. **B, B'.** Images showing a peak of Ase cell (red arrow) neighboring a strong Dl expressing cell (green arrow) at the NE border. **C.** A peak of Ase cell (arrow) exhibits strong E(spl)mγ-GFP labeling. **D, E.** The expression of two *Su(H)-LacZ* reporters is strong in NBs (Mira+), weak in the peak of Ase cells and L'sc+ cells, and practically absent in NE cells. **F.** Schematic representation of the expression profiles of Ase, Dl, E(spl)mγ (Notch signal activity), L'sc and Su(H) along the distinct cellular transition stages based on the data presented. The nuclear and cell wall colors correspond to Ase, Dl, E(spl)mγ, L'sc and Su(H) as indicated.

7 brains), suggesting that Notch signaling promotes *ase* expression during the NE-NB transition. This possibility is also supported by the observation that the down-regulation of *Notch* with Dl-DN appears to repress *ase* expression (Fig 4D and 4E; S5C–S5E Fig S1 Data). Furthermore, the peak of Ase expression was severely suppressed when Notch activity was blocked in transitioning cells by expressing Dl-DN under the control of the *c820-Gal4* driver (Fig 5E–5H; 12/12 brains). Nevertheless, activating Notch signaling inside the NE territory with clones expressing the Notch intracellular domain (N_{ICD}) did not induce Ase expression inside the clones while the cells remain expressing NE markers (Fig 5I and 5I'; 63/65 clones, 24/24 brains). Remarkably, when we induced the expression of Dl in small *Dl GoF* clones located in

the NE, ectopic Ase-expressing cells were observed at the outer border of these clones, at the edge of the NE but not inside it (Fig 5J and 5J'; 15/20 clones). Hence, the activation of Notch signaling by Dl at the NE border appears to induce ectopic peak of Ase cells, reminiscent of the effect of L'sc GoF clones described above (Fig 5C and 5D). By contrast, although *Dl* expressing clones located in the NB region contained *ase* expressing cells (Fig 5J,J'), $N_{ICD}$ clones located in the NB area of the OPC lack Ase expression (Fig 5K and 5K'; 82/87 clones). Conversely, the downregulation of Notch in NBs with Dl-DN driven by Ase-Gal 4 in medulla NBs produced an increase in Ase labeling intensity compared to control NBs (Fig 5L–5N: S1 Data, 6/6 brains). Together, these findings indicate that Delta-Notch activity is necessary and sufficient to induce *ase* expression at the medial but not in the lateral side of the NE-NB transition zone while Notch signaling appears to be repressive in NBs.

Given that the peak of Ase expression appears to be required for a correct NE-NB transition, we asked how the expression of *ase* is down-regulated in NBs that also exhibit strong Notch activity (Fig 6C and 6F). We wondered if Dpn could be involved in this downregulation of Ase since Dpn seems to be upregulated in the OPC NBs immediately after the peak of Ase expression, and Dpn consensus sites have been found in the *ase* gene [50]. Furthermore, we previously found that Dpn overexpression diminished *ase* expression substantially in NBs [77]. The *c820-Gal4* line was used to express *Dpn* RNAi and while this extensively depleted Dpn in NBs, it did not enhance *ase* expression (S7 Fig). Hence, Dpn alone does not seem to be required to repress *ase* in NBs. Nevertheless, we cannot rule out functional redundancy with other bHLH genes that might substitute Dpn to dampen *ase* expression, as reported in other cell contexts [81].

As an alternative candidate, we considered Su(H), the main mediator of Notch signaling in *Dm* [57,82] that follows an interesting expression pattern during the NE-NB transition. Thus, two independent Su(H)-Lac Z reporters showed that expression is apparently absent from NE cells, it is weak in transitioning cells and strong in NBs (Fig 6D–6F). Consistent with this expression pattern and Su(H) being a repressor of *ase*, expressing Su(H) in NE and transitioning cells under the *c820-Gal4* driver caused a widespread reduction of Ase expression and a fall in the number of peak of Ase cells (Fig 7A–7C; 10/10 brains; S1 Data). Conversely, expressing the Su(H) RNAi under the control of the *c820-Gal4* driver enhanced Ase expression in NBs (Fig 7D and 7E; 19/23 brains). These data support the idea that Su(H) is necessary and sufficient to downregulate *ase* expression in NBs.

## L'sc expression is promoted by Notch signaling and it is down-regulated by Su(H) at the NE-NB transition

The current NE-NB transition model proposes that the activation of Notch in the NE represses *l'sc* expression to prevent a premature switch to the NB fate [10,28]. Strikingly, and in clear contrast to this premise, we found that the majority of the most lateral L'sc expressing cells exhibited intense Notch signaling, as detected with the E(spl)-mγ reporter (69%: Fig 6A; 6/6 brains). As these are the cells in which *l'sc* expression commences, we wondered whether Notch signaling at the lateral NE border could actually trigger *l'sc* expression. Indeed, we found that L'sc expression was induced extensively in $N_{ICD}$ clones originated in the NE territory (Fig 8A and 8A'; 69/90 clones, 20/20 brains). Noteworthy, L'sc was always found inside these clones. Conversely, neither Notch RNAi expressing clones originating in the NE (Fig 8B and 8B'; 21/21 clones, 5/5 brains) nor the expression of Dl-DN in NE cells driven by *c855a–Gal4* (S8A and S8B Fig) induced L'sc expression. Finally, we encountered that Su(H) GoF controlled by the *c855a-Gal4* driver extensively suppressed the expression of L'sc (Fig 8C and 8D; 19/19 brains), whereas the Su(H) RNAi under the *c820-Gal4* driver led to the appearance of

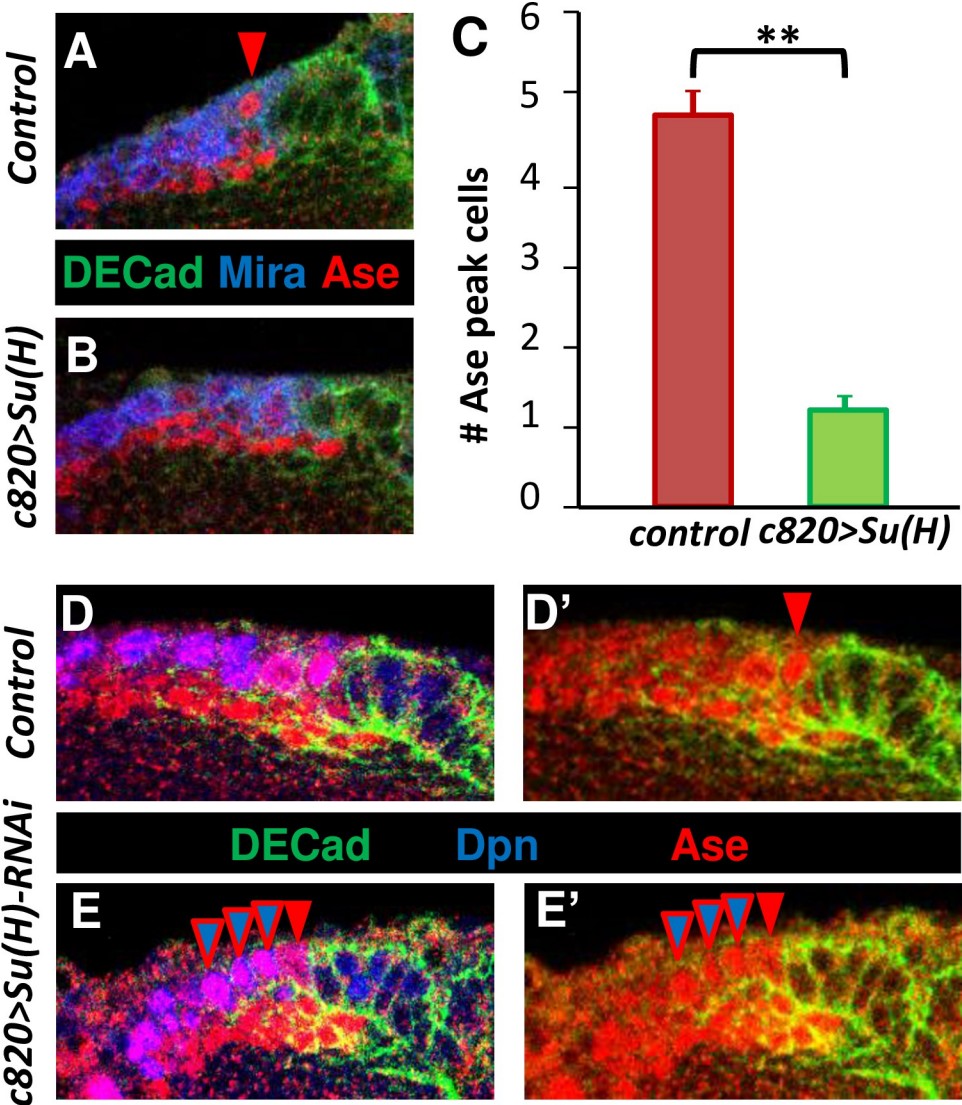

**Fig 7. Effects of Su(H) LoF and GoF on Ase expression at the NE-NB transition. A, B**. Effect of *Su(H) GoF*. The OPC of a representative *c820>Su(H)* specimen lacks peak of Ase cells (arrowheads). **C**. Quantification of the number of peak of Ase cells along 20 µm of OPC Z axis in 10 larval brains of control (*c820-Gal4*) and *c820-Gal4/UAS-Su(H)* larvae after a 16h induction. Differences are statistically significant (Student t-Test, P<0.001). **D, E**. Effect of *Su(H)* downregulation. The OPC of a representative *c820>Su(H)-RNAi* sample exhibits numerous Dpn[+] cells with high expression of Ase (blue/read arrows), similar to the peak of Ase cell (read arrow) compared to the control specimen.

ectopic L'sc[+] cells in the medial side out of the NE (Fig 8E and 8F; 14/15 brains). Together these data show that Notch GoF in the NE and Su(H) GoF in the transition cells are sufficient to sequentially promote and repress *l'sc* expression, respectively.

## Discussion

### Ase plays the dominant proneural role in the NE-NB transition

Among the key players regulating events in neurogenesis, the proneural bHLH TFs deserve special attention as they fulfil major evolutionary conserved roles. Studies in *Dm* and vertebrates have shown that these TFs are necessary and sufficient to initiate a developmental program that

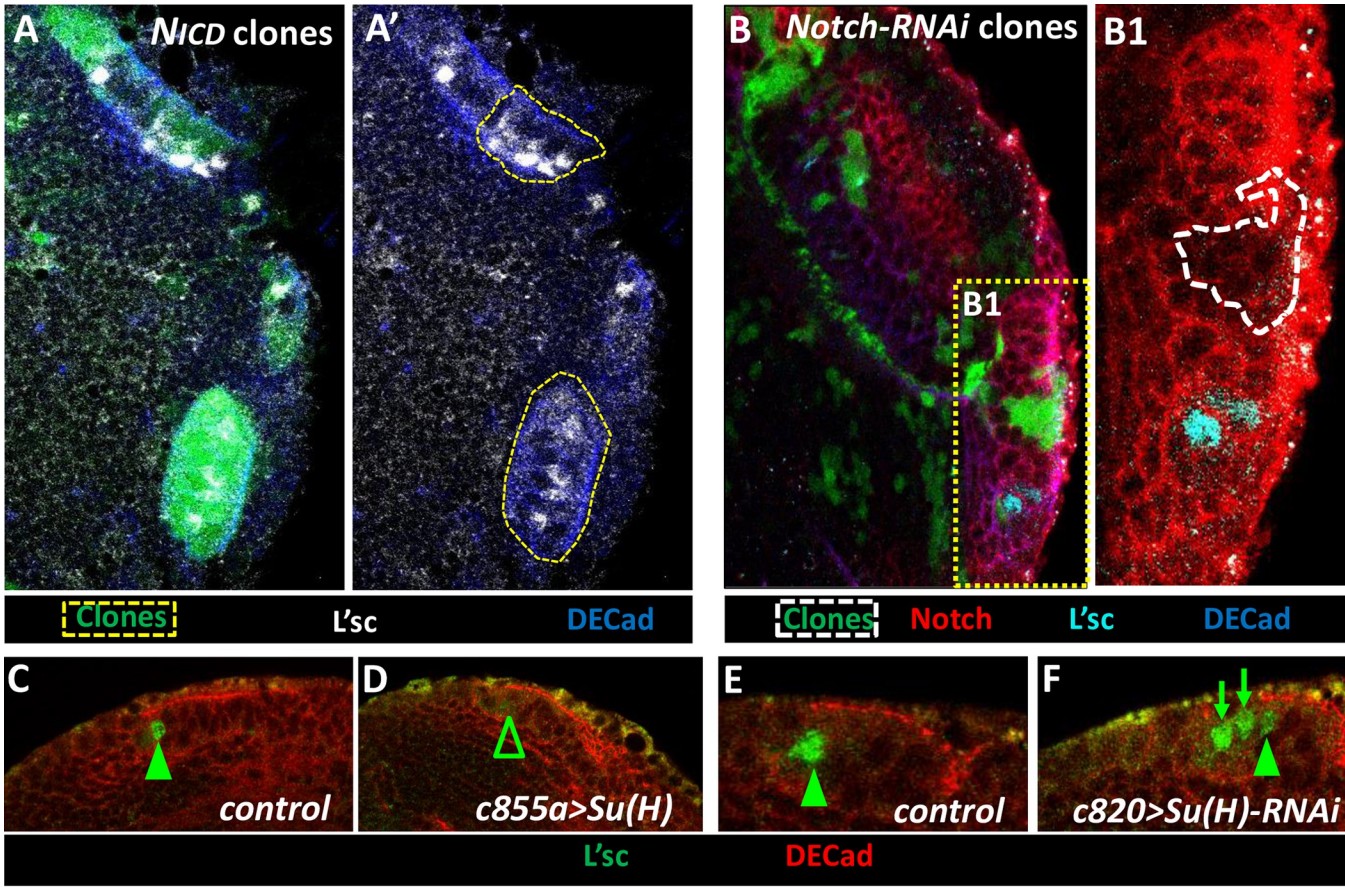

**Fig 8. Effects of Notch and Su(H) GoF and LoF on L'sc expression. A, B.** Clonal analysis of the effect of Notch on L'sc expression. **A, A'.** Two clones misexpressing the N$_{ICD}$ domain in the OPC NE show ectopic expression of L'sc. **B, B1.** By the contrary, a similarly localized Notch-RNAi clon with severely decreased expression of Notch, lacks L'sc expression. **C-F.** Effect of Su(H) on L'sc expression. Misexpression of Su(H) in the NE (*c855a>UAS Su(H)*) decreases L'sc expression in the transition cells (**D,** arrowhead) whereas the downregulation of Su(H) in transition cells (*c820>Su(H)-RNAi*) yields ectopic L'sc expressing cells (**F,** arrows) compared to their respective controls (**C, E**).

generates NPs committed to produce neuronal and glial lineages [42–44,52,54,83,84]. In *Dm*, the proneural members of the AS-C, *ac*, *l'sc* and *sc* play a key role in driving epidermal cells towards a neural fate. Thus, they are expressed in clusters of ectodermal cells (proneural clusters) and they promote the formation of sensory organ precursors (SOPs) in the embryonic and adult PNS, and of NBs in the embryonic CNS [42–44]. Initial studies indicated that *ase*, the fourth member of the AS-C, was functionally different since it is not expressed in the ectoderm but rather in SOPs and their lineages, where it persists longer than the proneural AS-C factors [46,47,85,86]. Moreover, *ase* deletion causes abnormal differentiation of sensory organs [51]. Accordingly, *ase* seems to be a neural precursor gene rather than a proneural gene. Nevertheless, Ase plays a proneural role in the generation of some wing margin bristles [51] and it also displays proneural potential since its ectopic expression is capable of initiating sense organ fate [46,51].

In the developing larval CNS, *ase* was found to be expressed by NBs and their progeny [46,47], which in principle also points to a role as a neural precursor gene. Indeed, Ase has been used extensively as a Type I NB marker. Nevertheless, we found previously that *ase* was transiently upregulated in GCs generated by Type I and OPC NBs, playing a key role in the mechanisms that induce the cell cycle exit and terminal differentiation of these neuronal

precursors [48]. Remarkably, this function very much resembles the classic role played by the AS-C vertebrate orthologues Ascl1/Mash1 and Neurogenins in promoting cell cycle arrest and differentiation of neuronal precursors [52,53,84].

Two types of intermediate NPs have been described as the NE-NB transition progresses: PI progenitors are defined by the expression of NE markers and strong Notch activity, whereas PII progenitors, while still expressing NE markers, are distinguished by strong *l'sc* expression and weak (or no) Notch activity [28,32]. We show here that *ase* is upregulated just after PII progenitors. Thus, the peak of Ase expression identifies a third step (PIII) in this transition since these strongly expressing Ase cells have lost their NE makers and L'sc expression but they do not consistently express NB markers yet and they receive a strong Notch signal (Fig 6F). Most importantly, our results demonstrate that the upregulation of *ase* is necessary and sufficient to promote the NE to NB transition, providing the first direct evidence for a pro-neural role of Ase in CNS neurogenesis. Noteworthy, the expression of Ase has also been associated to the transition from immature to mature secondary NBs in type II NB lineages at the larval CB. However, Ase does not seems to be required to promote this transition since the LoF of *ase* does not produce alterations in the lineage [87].

Our data also show that *l'sc GoF* in OPC NE cells is not sufficient to promote the transition to NBs under the same conditions in which Ase induces a complete transformation of NE cells into NBs. Intriguingly, *l'sc GoF* was previously shown to induce extensive neurogenesis in the OL [31,33]. We believe this could be explained by the initial induction of *l'sc GoF* from embryonic and very early larval stages, developmental periods in which L'sc appears to be involved in the formation of the optic placode [17,88]. By contrast, to avoid interfering with the initial process of OPC NE development, here we induced *l'sc* from the mid-third instar stage. This might also be the reason why it was previously found that L'sc induces anomalous delamination of NPs from the NE inside the OL [33], something that resembles the regular transformation of a subset of IPC NE cells into migratory NPs, a process in which L'sc is involved [89]. It should also be noted that the *NP6099 Gal4* driver that controlled *l'sc GoF* in this earlier study [31] seems to drive its expression in LPCs [90] rather than OPC NE cells during late larval stages at least. In contrast, here we used the *c855a-Gal4* line to drive *l'sc GoF* in OPC NE cells and remarkably, we detected an exiguous proneural effect in the medulla NE but a consistent one in the lamina NE.

Our results are also consistent with the classic actions of Ascl1 at sequential stages of vertebrate CNS neurogenesis, specifying NP subtypes as well as cell cycle exit and the terminal differentiation of neurons [52,54,84,91,92]. Nevertheless, we find no indications that Ase fulfils a role in promoting NP proliferation in the OPC, as described in a subset of IPC NBs [89], or along the line of Ascl1 activation of positive cell cycle regulators in the mouse telencephalon [53]. Actually, the peak of Ase expression during the NE-NB transition possibly coincides with a prolonged cell cycle of transitioning cells [34,76], which in principle points to an anti-proliferative activity of Ase, as already described in the NB progeny [48,49]. In that sense, the downregulation of *ase* after its peak of expression in PIIIs could allow NBs to continue cycling. Thus, it would be interesting to analyze the possible role played by Ase in the regulation of the cell cycle during the NE-NB transition in depth.

In summary, the present results together with our previous data [48,77] demonstrate that Ase is the proneural TF with the dominant role regulating the program of neurogenesis of the medulla OPC by promoting the differentiation of NPs in successive phases: first, from proliferative (NE) to self-renewing (NBs) NPs; and afterwards, the cell cycle exit and terminal differentiation of neuronal precursors (GCs).

## Non-conventional integration of Delta-Notch signaling, L'sc, Ase, and Su (H) sequential activities ensures a timely NE-NB transition

Elucidating the genetic programs and molecular mechanisms that control neurogenesis has been a major focus of research over the past three decades [2,24,25,93–95]. Consequently, many genes and individual mechanisms that operate at different stages of neurogenesis have been identified. Nevertheless, how these mechanisms are integrated to coordinate the whole neurogenic process remains poorly defined in many tissues.

The integration of proneural factors and Notch signaling is pivotal in the regulation of neurogenesis. The specification of NBs in the central brain and ventral nerve cord of the *Dm* embryo is the model of reference for this integration. In brief, these NBs are specified in a two-step process. First, the expression of AS–C proneural factors in clusters of neuroectodermal cells ("proneural clusters") make these cells competent to become NBs. In a second step, one of the cells in each cluster is specified as a NB through lateral inhibition, a regulatory loop that is mediated by Notch signaling through the upregulation of Dl expression by the proneural factor and the down regulation of the proneural gene by the Notch signal. As a consequence, the cell with increasing level of the proneural factor (and hence, Dl) will repress the proneural factor (and Dl) in the neighboring cells. This will drive a feedback loop that increases proneural expression at the cell that will be specified as a NB, while its neighbors will become epidermal cells [45,82,96].

The NE-NB transition is a complex process regulated by multiple signaling pathways (JAK/STAT, EGFR, Fat-Hippo and Notch), although how they are coordinated is only just beginning to be understood [10,15,28]. Among them, proneural factors and Notch signaling play critical roles in this transition. For instance, Notch signaling seems to be involved in several sequential steps in this transition and initially, it is fundamental to maintain NE cell fate and proliferation [29,32–35,76,97]. Similarly, the Notch pathway plays a key role in maintaining NPs undifferentiated during the early neurogenic phases in the developing vertebrate CNS [98–100]. After the transition, Notch signaling seems to be also essential for NB proliferation [81,101]. However, it is less clear how Notch signaling acts at the intermediate stages between NE cells and NBs and how this is coupled to the actions of proneural factors.

Notch activity is very dynamic during the NE-NB transition. First, Notch is weakly activated across most of the NE due to the input of the Serrate (Ser) ligand from the surrounding glia [97], and this activation is likely involved in maintaining the NE fate. Secondly, Notch activity becomes very strong at the most medial edge of the NE (PI transition cells), where in addition to the glial Ser input, an intense signal is received from the strong Dl expressing PII transitioning cells (Fig 9C). Other additional signaling acting at this stage, such as EGFR that has been shown to promote Dl expression [36], could also contribute to this initial regulation. (Fig 9A and 9B). Subsequently, the Notch pathway is repressed in PII cells, possibly through the combined activity of L'sc [32,33,80] and the negative feedback loop with Dl [34,76] that provokes degradation of the Notch protein by Dl cis-inhibition [78]. Finally, Notch signaling is also strongly activated in the adjacent PIII transitioning cells by the highly expressed Dl of PIIs (Fig 9C). In accordance with this complex Notch signaling pattern, *Dl LoF* and *GoF* clones produce both inhibition and acceleration of NB formation [34]. Interestingly, *Notch LoF* clones produced a rather different phenotype depending on their medio-lateral position in the OPC [76]. These data strongly suggest that Notch signaling can produce opposing effects along the NE-NB transition axis.

Remarkably, we show here that L'sc expression is promoted by Notch signaling. This action appears to work cell-autonomously since L'sc was always induced inside $N_{ICD}$ clones. This is a non-conventional effect since, as discussed above, Notch signaling represses proneural gene expression in classic neurogenic CNS models [52,82,96].

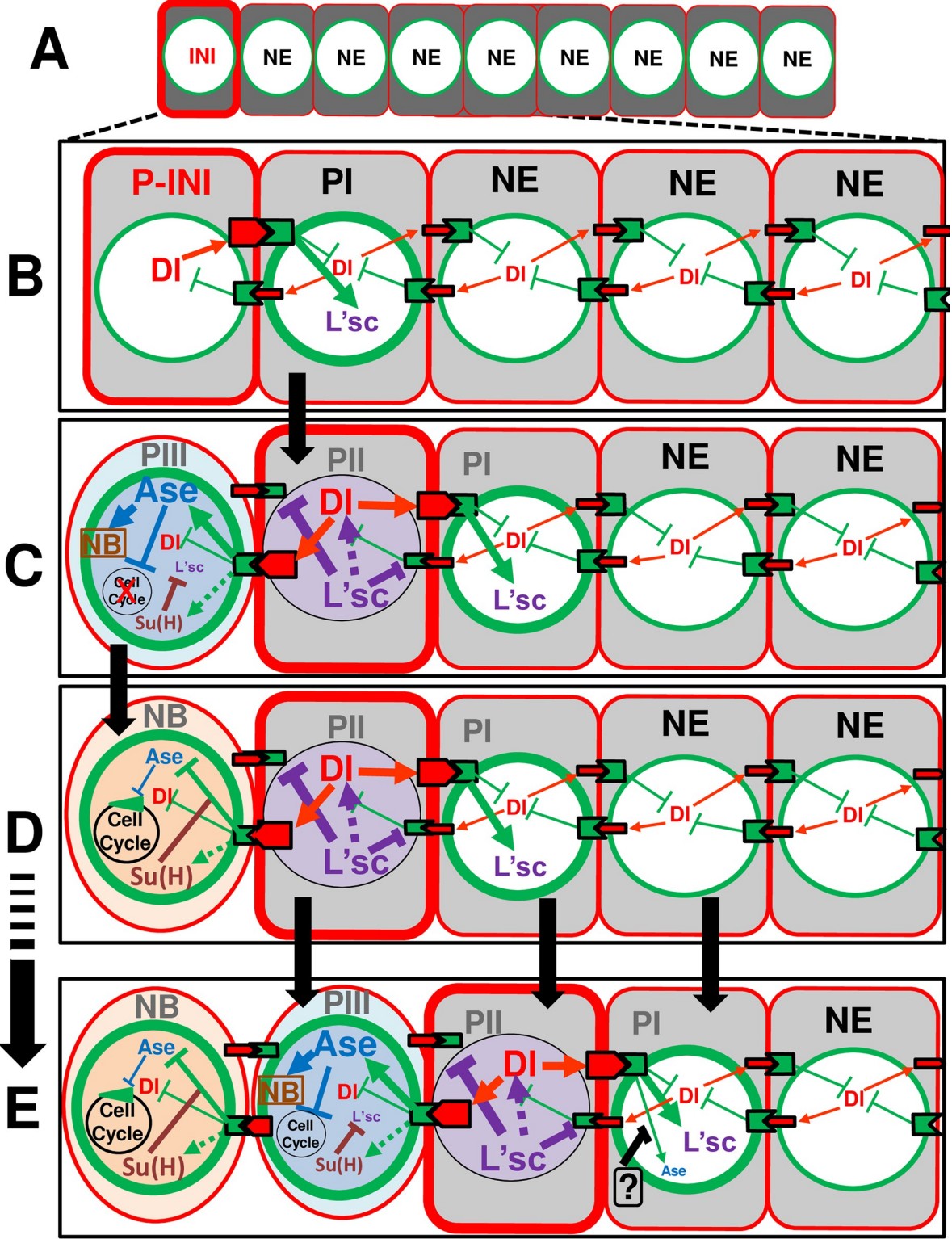

**Fig 9. Working model: integration of Ase proneural function with Delta-Notch signaling, L'sc, and Su(H) sequential actions promotes the NE-NB transition.** Working model with the interactions of Ase, Dl, L'sc, Notch activity and Su(H) facilitating the stepwise transition of NE cells into NBs. **A**. We hypothesize that transition is initiated at the medial side of the NE by an increase in Dl expression (INI). **B**. This induces a strong Delta-Notch signal in the neighboring cell (PI), which promotes L'sc expression. As discussed elsewhere, other signaling pathways (such as EGFR) can be involved in this initial step. **C**. The upregulation of L'sc represses Notch allowing Dl expression. Thus, this

cell becomes a PII progenitor. In turn, the high Dl expression in PII drives strong Notch signaling on both sides cells. On one hand, this promotes the expression of Ase (and possibly Su(H)), transforming that cell into a PIII. On the other side, Notch signaling promotes L'sc and hence, generates a new PI. In the PIII, Su(H) represses L'sc while Ase arrests the cell cycle and promotes NB fate genes transforming this cell into a NB (**C-D**) where the increased expression of Su(H) turns Notch signaling into repressive for Ase, allowing the cell cycle to proceed. **E**. Consecutive rounds of transformations as those described in A-D yield one further step in the wave. The "**?**" element in PI refers to an unknown factor/signal that precludes the premature expression of Ase by Delta-Notch signaling in PI.

Remarkably, the extensive presence of L'sc⁺ NE cells within $N^{act}$ (*Notch GoF*) clones was previously interpreted as a delay in the NE/NB transition [30,35]. However, we think that this reveals the capacity of Delta-Notch activity at the medial NE edge cell (PI) to trigger L'sc expression and hence, to drive the next step in the transition (i.e. the transformation of PI into PII progenitors) (Fig 9B–9C). Consistent with this initial role for Delta-Notch signaling in driving the transition, it has been concluded that the first peak of Notch activity is responsible for regulating the propagation speed of the proneural wave [37].

It has been clearly shown that EGFR signaling is activated in the most medial NE and transitional cells of the OPC and, accordingly, it was proposed that this signal induces L'sc expression [32]. Nevertheless, the ectopic activation of EGFR signaling in clones resulted in a lateral shift of the L'sc expression peak respect to the surrounding *wt* tissue [32] but not in the upregulation of *l'sc* within most, if not all, the clone area, as we have seen by the strong activation of Notch signaling with $N_{ICD}$. Interestingly, overexpression of $N^{act}$ resulted in the expression of PntP1, a major transducer of EGFR signaling [102], in NE cells [32,36]. Furthermore, the overexpression of $N^{act}$, in *pnt LoF* clones covering the NE-NB transition zone resulted in the lack of L'sc expression [32]. These data suggest that the upregulation of *l'sc* by Notch signaling may require the priming activation of EGFR signaling. Noteworthy, the coactivation of Notch and EGFR signaling in clones seems to produce the same effect in the transition as the activation of Notch alone since the cells remain expressing L'sc and do not progress to NB differentiation [32]. Similarly, we have here shown that L'sc is not sufficient to drive further the transition cell-autonomously. Thus, this process stalls at the PII stage. This may indicate that L'sc needs to be downregulated for the transition to progress further. As we discuss here below, our results suggest that Su(H) is involved in this downregulation (Fig 9C).

Our data also show that Delta-Notch signaling is necessary and sufficient to promote *ase* expression in PIII progenitors. This is a second non-conventional effect of the Notch pathway on a proneural factor in the NE-NB transition. Furthermore, in contrast to classic neurogenic CNS models where NP production is associated with decreased Notch activity [52,82], the peak of Ase (PIII) cells receive strong Notch signaling (Fig 9C and 9D). Interestingly, it has been proposed that this second peak of Notch activity (i.e. behind the proneural wave) controls neurogenesis [78].

It is well known that the effects of Notch signaling depend very much on the cell context [103–107]. Remarkably, two sequential actions of Delta-Notch signaling arise in the caudorostral wave of neurogenesis in the prospective spinal cord of the chick. The first of these maintains the caudal neural stem cells pool and the second mediates the transition of proliferating NPs into neurogenic NPs in the transition zone [108]. In this regard, this wave of spinal cord neurogenesis resembles the proneural wave in the *Dm* OPC. Multiple effects of Notch signaling have been also described in the development of the *Dm* eye where, as in the NE-NB transition, neural differentiation occurs progressively in a wave [109]. In this system, Notch signaling plays sequential roles in promoting and inhibiting neural determination. These opposing effects first involve promotion of the proneural gene *atonal (ato)* ahead of the morphogenetic furrow in response to Hh signaling and subsequently, the restriction of *ato* expression to R8 photoreceptor precursors behind the furrow through classic Delta-Notch lateral

inhibition [110–116]. Thus, similarly to what we have found for L'sc in the OPC, Notch signaling promotes the expression of the proneural factor Ato in a "pre-proneural" cell context to initiate neural eye differentiation.

During the classic process of lateral inhibition, the Notch intracellular domain converts the nuclear Su(H) factor from a transcriptional repressor into an activator, leading to the expression of E(spl) factors. Notwithstanding the fact that Su(H) is the major mediator of Notch signaling in *Dm* cell fate decisions [82,117], there is also experimental evidence that Notch signals can be transmitted independently of Su(H) [118–123]. Remarkably, the proneural enhancement and lateral inhibition sequential actions driven by Notch in the eye differs from the role of Su(H) [119,120]. Our data, based on two independent Su(H)-Lac Z reporters suggest that Su(H) is differentially expressed during the NE-NB transition with a possible increasing upregulation from PII transitioning cells towards NBs (Fig 6F). In agreement with this expression pattern, it has been shown the apparent lack of capacity of Notch signals driven by Ser to induce Su(H) expression in the NE, in contrast to the positive effect of the Delta driven Notch signal in the transition zone [97]. Nevertheless, like Ase, it is unclear why Su(H) is not upregulated in PI cells that also receive a strong Dl input.

In contrast, it has been reported that an anti-Su(H) antiserum exhibited a widespread immunostaining in the larval brain including most, if not all, OL cells [35] although no genetic controls were used to ensure the specificity of that expression pattern. In any case, the fact that *Su(H) LoF* clones located in the medial side of the NE showed alterations in the NE-NB transition [32,35], indicates that Su(H) is possibly required before the proneural wave for a proper transition. Independently of this, we have shown here that Su(H) is necessary and sufficient to downregulate *l'sc* and *ase* expression peaks (Fig 9C and 9D), suggesting its involvement in ending the proneural wave. The facts that we have found that Notch signaling promotes Ase in the transition but represses it in NBs suggest that Su(H) may help to switch Notch signaling from promoting to repressing *ase* expression at these transitioning stages (Fig 9C and 9D).

Notably, our Dl and *l'sc* GoF data indicate that *ase* expression is repressed tightly in the NE by an unknown mechanism that possibly serves to prevent its premature upregulation by Notch signaling in the lateral side, ensuring in this manner the medial to lateral direction of the neurogenic wave. Thus, despite the strong Notch signals driven by Dl into PI cells, *ase* is only activated at PIII cells (Fig 9D). This might require the expression of L'sc that promotes the expression of Dl in the previous PII stage since we found that its suppression in these cells precludes the subsequent peak of Ase expression. Additionally, the upregulation of Dl-Notch signal in PII can also promote Su(H) that in turn can converts the positive effect of Notch signal for Ase expression at the transitioning PIII into a repressive one in NBs (Fig 9C and 9D).

Despite its crucial role, the LoF of *ase* does not fully block the NE-NB transition. Thus, the whole transition is a very robust process that relies on partially redundant mechanisms. In that regard, and as previously reported [29,32–35,76], we found that Notch LoF causes transformation of NE cells into NBs, although this requires long time. This could be part of a parallel mechanism, as we have seen that the downregulation of Notch is unable to promote *l'sc* or *ase* expression in the short term. Moreover, this seems to occur independently of Ase since we have observed that NB markers were induced earlier than Ase after the down-regulation of Notch and it has been reported that Ase was not upregulated in a large part of *Notch LoF* clones 2.5 days after clonal induction [76]. Remarkably, we have here shown that *Notch LoF* induces the NE-NB transition at a slower rate than *ase GoF*. Noteworthy, *Notch LoF* clones located in the lateral region of the OPC NE (i.e. far from the transition zone) did not induce NBs precociously and continued to exhibit NE characteristics, as opposed to those clones located at its medial edge [35, 76], which are close to the transition zone. Together, these data strongly suggest that the down-regulation of Notch rather than being a final instructive action

promoting NB fate, it is an intermediate permissive or priming step that requires additional signaling to proceed. The fact that in double *Notch + EGFR signaling LoF* clones cells remain in the NE stage supports this idea [32].

Although the *ase* gene promoter contains binding sites for AS-C/daughterless heterodimers [86], and *ase* lies downstream of proneural genes in PNS neurogenesis[46], our data indicate that the action of L'sc on *ase* expression in the OPC is not direct but rather, it is mediated by the non-cell autonomous activation of Notch signaling in the adjacent cell. This could be one of the main functions of L'sc in this context. In addition, L'sc seems to also be involved in promoting the transition from the PI to the PII progenitor stage [80]. Through these synchronized activities, L'sc is crucial to control the timing of the NE-NB transition.

As discussed before, in addition to the cell autonomous promotion of neurogenesis, proneural factors activate Notch signaling non-cell autonomously in adjacent progenitors by promoting the expression of Notch ligands [43,52]. Our results support the idea that these two parallel proneural functions have been split between L'sc and Ase during the NE-NB transition. Thus, L'sc appears to be responsible for mediating the activation of Notch in the adjacent PIII progenitors, while Ase seems to be the final effector that promotes the differentiation of these cells into NBs. Interestingly, the change in the expression of two proneural proteins (Ase to Ato) regulates another key NP transition in CNS neurogenesis, the switch of a subset of IPC NBs from asymmetric divisions to transient amplification divisions [124].

Together, our data reveal clear differences in the way proneural and Notch signaling actions are integrated in the NE-NB transition relative to classical models of neurogenesis. We think that the main reason underlying these differences resides in the divergent strategies adapted to different cell patterns of neurogenesis: "salt-and-pepper" (embryonic neuroectoderm) as opposed to a neurogenic wave across the tissue (eye and OPC). Thus, it would be interesting to investigate the involvement of other signaling pathways in the differential integration of proneural and Notch signaling actions in these experimental models.

In summary, our results reveal a novel and crucial proneural function of Ase in CNS neurogenesis, which is non-conventionally integrated with Delta-Notch signaling, L'sc, and Su(H) timely activities in order to promote the progressive transformation of NE cells into NBs. Thus, our data helps to reformulate the working model of the NE-NB transition and opens the question about the evolutionary conservation of these regulatory mechanisms in vertebrate CNS regions where the transition from proliferating to neurogenic NPs follows a neurogenic wave.

## Supporting information

**S1 Fig. Expression patterns of Gal4 drivers at the NE-NB transition.** Surface (**A1, B1**) and deep layer confocal sections (**A2, B2**) showing GFP expression in the OPC of *c820-Gal4/UAS-dGFP* and *c855a-Gal4/UAS-dGFP* larval brains. In the *c820>GFP* sample, GFP is present in the transition cell (L'sc[+]) and in medial NE cells, while the *c855a>GFP* specimen exhibits strong GFP labeling in NE cells but weak or absent in transition cells.
(PDF)

**S2 Fig. Effect of temperature on the expression of GFP, Ase, and L'sc induced by *c855a-Gal4*.** GFP expression in the OPC (deep layer) of *c855a-Gal4/UAS-dGFP* larval brains incubated at 17°C all along the larval developmental time (**A**) or at 17°C until mid third instar stage followed by 24h at 30°C (**B**) (see Materials and Methods). **C,D**, Confocal images taken close to the surface of the OPC of control and *c855a-Gal4/UAS-ase* larval brains incubated at 17°C. Only 3 out of the 7 *c855a-Gal4/UAS-ase* analyzed brains presented a very subtle phenotype consisting on having 1 Dpn+ cell (arrowhead) within the OPC NE. **E**. Deep layer confocal

image of the OPC of a *c855a-Gal4/UAS-L'sc* larval brain incubated at 30°C during the last 12h of development (see Materials and Methods). Note that the ectopic L'sc⁺ cells driven by *c855a-Gal4* in the NE (white elipse) have stronger labeling than the endogenous (transition) L'sc + cells (green elipse).
(PDF)

**S3 Fig. Alterations of the NE-NB transition by *ase RNAi* driven by *c820-Gal4*. A, B**. Deep layer confocal section of control and *c820 >ase-RNAi* samples. **A1, B1**. High magnification views of the framed areas showing that the *c820 >ase-RNAi* sample lacks peak of *ase* cells compared to the control (green arrowhead) and exhibits weakly labeled Ase+ cells co-expressing Mira and DECad (read/white arrowhead). **C**. Quantification of the number of peak of Ase cells along 20 μm of OPC Z axis in 10 larval brains of control (*c820-Gal4*) and *c820>ase-RNAi* larvae. Differences are statistically significant (Mann-Whitney Rank Sum Test, P<0.001) **D.** Quantification of Mira/Cad co-expressing cells in control and *c820>ase-RNAi* (13 OLs). Statistical significance was assessed with Mann-Whitney Rank Sum Test (P<0.001).
(PDF)

**S4 Fig. Differential effects of Ase and L'sc GoFs on the expression of Mira in the NE.** Confocal images taken close to the surface (**A,C**) or in deep layers (**B,D**) of the OPC of control (*c855a-Gal4*) and *c855a-Gal4/UAS ase* larval brains after a 8h induction. Note the presence of Mira⁺ cells intermingled in the NE (yellow arrowheads) of the *c855>ase* specimen compared to the control. **E-H**. Equivalent images taken from control (*c855a Gal4*) and *c855a Gal4/UAS l'sc* larval brains after a 12 h induction. Note that despite the large number of L'sc⁺ cells (green arrowheads) inside the medulla NE (left side) of the *c855>ase* specimen there are no Mira + cells on it. In contrast, there are several ectopic Mira+ cells in the lamina NE (right side, orange arrowheads).
(PDF)

**S5 Fig. L'sc misexpression and Notch down regulation do not induce Ase expression in the NE.** Confocal images taken close to the surface (**A,B**) of the OPC of control (*c855a Gal4*) and *c855a-Gal4/UAS-Dl-DN* larval brains after a 9 h induction. Note the lower Notch labeling in medial edge of the NE in the control specimen (B, white dotted line area) and the fall of Notch expression in the whole NE (green dotted line area) of the *c855a>Dl-DN* sample (B'), **C,D.** Confocal images taken in deep layers of the OPC of control (*c855a-Gal4*) and *c855a-Gal4/UAS-Dl-DN* larval brains after a 12 h induction. Note that in the *c855a>Dl-DN* sample, the first cell medial to the NE exhibits strong Mira and weak Ase labeling (green arrow) in contrast to the stoong peak of Ase expression in the control sample (red arrow). **E** Quantification of the number of peak of Ase cells along 20 μm of OPC Z axis in control (*c855a-Gal4*) and *c855a-Gal4/UAS-Dl-DN* larval brains after a 8h induction. Differences are statistically significant (Mann-Whitney Rank Sum Test, P = 0.002). **F,G.** Confocal images taken close to the surface of the OPC of control and, *c855a-Gal4/UAS-L'sc* larval brains after a 12h induction. The NE of the *c855a>L'sc* specimen is almost identical to the control sample except for the presence of a single Ase+ cell (arrow).
(PDF)

**S6 Fig. Timing of Notch protein downregulation induced by *Notch RNAi* driven by *c855a-Gal4*. Effects on the NE-NB transition.** Close to surface confocal image of the OPC of control and *c855a>N RNAi* samples after induction for 9 and 36h. Notice that after 9h induction there is no apparent decrease in Notch labeling compared to the control (**A', B'**). In contrast, after 36h induction Notch labeling is very weak (**C'**). Numerous Dpn/DECad co-expressing cells

(arrows) can be detected in the *c855a>N RNAi* sample after 36h induction (**C**)
(PDF)

**S7 Fig. The downregulation of *Dpn* in NBs does not modify Ase expression.** Confocal images taken at deep layers of the OPC of control (*c820-Gal4*) and *c820-Gal4/UAS-Dpn-RNAi* larval brains after a 24h induction. Note that despite the complete suppression of Dpn immunostaining there is no increase in Ase labeling in the NBs of the *c820>Dpn-RNAi* relative to the NBs of the control sample and to its own Ase peak cell (arrowhead).
(PDF)

**S8 Fig. The downregulation of Notch in the NE does not induce L'sc expression.** Confocal images taken at equivalent deep layers in the OPC of control (*c855a-Gal4*) and *c855a-Gal4/ UAS-Dl-DN* larvae after a 36h induction. Notice that there are not additional L'sc+ cells in the *c855a-Gal4>-Dl-DN* sample compared to the control.
(PDF)

**S1 Data. Numerical data and summary statistics of graphs.** Excel file containing the numerical data and summary statistics used to generate the graphs displayed in the corresponding figures. The data is organized in independent spreadsheets for each figure graph, as indicated.
(XLSX)

## Acknowledgments

We are very grateful to S. Campuzano, Jose F. de Celis, and Juan A. Sanchez-Alcañiz for useful suggestions and critical reading of the manuscript. We are indebted to A. Baonza, H. Bellen, S. Bray, S. Campuzano, A. Carmena, C.Q. Doe, M. Dominguez, C. Estella, C. Gonzalez, Y-N. Jan, A. Jarman, F. Matsuzaki, I. Saelecker, M. Sato, S. Selleck, J. Skeath, G. Struhl, H. Vaessin, the Vienna Drosophila Resource Centre (VDRC), the Bloomington Drosophila Stock Center (BDSC), and the Developmental Studies Hybridoma Bank (DSHB) for providing us with flies, antisera and other molecular tools.

## Author Contributions

**Conceptualization:** Francisco J. Tejedor.

**Formal analysis:** Mercedes Martin, Francisco Gutierrez-Avino, Mirja N. Shaikh, Francisco J. Tejedor.

**Funding acquisition:** Francisco J. Tejedor.

**Investigation:** Mercedes Martin, Francisco Gutierrez-Avino, Mirja N. Shaikh, Francisco J. Tejedor.

**Methodology:** Francisco J. Tejedor.

**Project administration:** Francisco J. Tejedor.

**Supervision:** Francisco J. Tejedor.

**Validation:** Mercedes Martin.

**Writing – original draft:** Francisco J. Tejedor.

**Writing – review & editing:** Mercedes Martin, Francisco J. Tejedor.

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
