## [Decision Letter · Decision Letter 0]

8 May 2023

Dear Dr Tejedor,

Thank you very much for submitting your Research Article entitled 'A novel proneural function of Asense is integrated with the sequential actions of Delta-Notch, L’sc and Su(H) to promote the Neuroepithelial to Neuroblast transition' to PLOS Genetics.

Your manuscript has now been seen by 3 reviewers with expertise in the field of Drosophila neurogenesis. The reviewers agree that the work contains important new insights into the mechanisms of neurogenesis and in particular into the interactions between Notch signaling and proneural genes. However, the reviewers also raise a number of concerns that I invite you to address. In particular, reviewers 2 and 3 point to a few technical issues that would be important to address - where possible and reasonable. In addition, reviewer 1 would like to see the work better discussed and integrated in the context of previous work on how Notch signaling interacts with other pathways that have not been examined in this work, and how some of apparent contradictions might be reconciled. Finally, all 3 reviewers suggest some caution in some of the interpretations, especially where no direct evidence is provided to support the conclusions.

Overall, I agree with the reviewers on the importance of making sure there are as few technical caveats as possible, and better integrating the main findings in the context of previous work, especially where the conclusions of this work differ from, or challenge, previous models. Should you wish to provide a revision plan for how you will respond to the concerns with new controls, extended analyses and textual changes, I would be happy to discuss that with you. Should you decide to revise the manuscript for further consideration here, your revisions should address the specific points made by each reviewer. We will also require a detailed list of your responses to the review comments and a description of the changes you have made in the manuscript.

If you decide to revise the manuscript for further consideration at PLOS Genetics, please aim to resubmit within the next 60 days, unless it will take extra time to address the concerns of the reviewers, in which case we would appreciate an expected resubmission date by email to plosgenetics@plos.org.

Yours sincerely,

Bassem A. Hassan

Guest Editor

PLOS Genetics

Gregory P. Copenhaver

Editor-in-Chief

PLOS Genetics

Reviewer's Responses to Questions

**Comments to the Authors:**

Reviewer #1: This paper from Martin et al. describes a new function of the transcription factor Asense (Ase) during the neurogenesis of the Drosophila optic lobe, particularly in the outer proliferation center (OPC). The main conclusions are that Ase works as a proneural gene promoting the NE-NB transition during OPC development and, on the other hand, there is a novel cell type in the transition between neuroepithelial cells and neuroblasts with the highest Ase expression that the authors called Ase peak cells (PIII in the discussion). Overall, the conclusions are interesting and generally supported by the data. However, some precision needs to be added in the description of the data, use of appropriate statistical analysis, and some additional experiments are required to be sure that the functions assigned to Ase are robust.

Major comments

1. In the time-restricted gene expression modification experiments, authors only use temperature, which is not ideal. Some strong GAL4 lines can also work at a low temperature. Why GAL80ts was not used for these experiments? For instance, it has been used in Egger et al. 2010.

2. The c820-GAL4 driver is not carefully described, some lower magnification pictures of the entire optic lobe will allow them to clearly demonstrate that this driver is specific to a subset of OPC cells.

3. Since most of the work intends to address ase function, using only one RNAi line is not ideal. Authors should try to test another RNAi line to be sure that part of the observed phenotype is not due to off-target effects.

4. In the same line, why not complement some of the RNAi experiments with MARCM clones or other loss of function ase clones? They already have a mutant of ase that could be recombined with an FRT chromosome.

5. Could the authors explain why although L(sc) is not able to promote by itself the NE-NB transition, it still can induce Ase expression in the transition zone?

6. Regarding Table I, what is the control condition? What is the statistical test used?

7. In Fig2G. What is the control? Just a control strain is not appropriate for an experiment in which a GAL4 is combined with an RNAi line. If the control is correct and three groups are going to be compared, a t-test is not the proper test to use, ANOVA should be used instead.

8. Regarding the Notch-dependent mechanism. How do the authors think that Notch promotes L´sc expression usually during the transition? Non-autonomous effect?

9. In suppl figure 4E authors use Mann-Whitney test to determine significance but in suppl Fig6, they use t-test instead. Please explain the rationale since is basically the same type of data. This explanation should be added to the methods section.

Minor comments

1. Please add line numbers, it makes the correction easier

2. Scale bars should be added to every figure

3. In the schematics, PI and PII cells are drawn with a rectangular shape in Figure 8, while PIII cells are round, while in Fig1 and Fig6, L´sc cell is round. Most literature agrees that PII cells are more like NE cells with high levels of DCad.

Reviewer #2: review uploaded

Reviewer #3: This study investigates the proneural role of various bHLH genes (l’sc, ase and Su(H)) downstream to Notch during the neuroepithelium-to-neuroblast (NE-to-NB) transition in the optic lobe of the Drosophila larval brain. In the larval optic lobe, a differentiation (or proneural) wave progressively converts the proliferative neuroepithelium into asymmetrically dividing neuroblasts. Various pathways have been shown to have key roles in the regulation of this wave such as the EGFR, JAK/STAT, Notch or ecdysone pathways. While this process has been actively investigated over the last decade, it is not completely understood. In particular, the dynamics of Notch signalling and the precise mechanisms by which Notch promotes or inhibits the proneural wave remain unclear. Here the authors investigate in details the sequential expression of Su(H), ase and l’sc along the progressive NE-to-NB conversion. They clearly show that the fine NE-to-NB transition involves 4 states defined by the sequential expression of E(spl), L’sc, Ase and Su(H). They propose that Ase can have a proneural role and that Su(H) is required in NBs to down-regulate Ase. Finally, they propose that these transitions are at least partly regulated by the Notch pathway in an unconventional manner, when compared to the classical lateral inhibition model.

The authors only focus on the Notch pathway and its interactions with bHLH genes, while other studies have shown that other pathways such as the EGFR and JAK/STAT can also interact with the Notch pathway to regulate its targets. Consequently, several of the conclusions somehow contradict the study of several labs (e.g. Tabata/Yasugi, Sato, Brand). There is therefore a need to clarify this situation in the discussion to demonstrate that this study brings a new and pertinent angle to our understanding of the mechanisms that drive the NE-NB transition.

More precisely:

- The authors mention that “GoF of ase is capable of inducing a more rapid and efficient NE-NB transition than the down-regulation of Notch”. It is unclear to me how relevant it is to compare the dynamics of a GOF and LOF experiments in such a complex system, especially when using mis-expression of a dominant negative form of Delta which may not lead to a complete knockout. Is the dynamics the same in Notch-RNAi conditions. Lastly, I am not sure to understand why this argument is necessary? Can you be more explicit about that?

- The authors proposes that L’sc expression is promoted by Notch signaling. This is contrary to what is observed during lateral inhibition where N represses L’sc. The authors interpret this result as a “non-conventional effect”, but they don’t provide any explanation on the underlying mechanism. This interpretation differs from various studies performed by Yasugi and Sato:

1: Sato M, Yasugi T, Minami Y, Miura T, Nagayama M. Notch-mediated lateral inhibition regulates proneural wave propagation when combined with EGF-mediated reaction diffusion. Proc Natl Acad Sci U S A. 2016 Aug 30;113(35):E5153-62. doi:10.1073/pnas.1602739113. Epub 2016 Aug 17. PMID: 27535937; PMCID: PMC5024646.

2: Yasugi T, Sugie A, Umetsu D, Tabata T. Coordinated sequential action of EGFR and Notch signaling pathways regulates proneural wave progression in the Drosophila optic lobe. Development. 2010 Oct;137(19):3193-203. doi:10.1242/dev.048058. Epub 2010 Aug 19. PMID:

These previous studies convincingly demonstrate using genetic experiments and mathematical modelling that l’sc is activated by EGFR signalling and failed to be activated by Notch signaling when EGFR signalling is down-regulated consistent with a pre-existing lateral-inhibition mechanism.

In the present manuscript, the authors do not take in account the potential role of EGFR to interpret their results, therefore it is unclear how this work integrates with the previous findings. It also makes it difficult to understand where the novelty is compared to these previous studies. This should be clarified.

Other points:

- Flybase indicates that the usual protein names of shg are E-Cadherin (E-Cad or ECad) or DE-Cad (for Drosophila ECad) but not DCad. It would be nice to stick to one of the previously used names.

- The ase-RNAi line does not seem to very efficiently knockdown Ase, and the

phenotype is hard to notice on the figure. It may be wise to test other RNAi lines and check whether phenotypes are more convincing.

- In some panels, the genotype is indicated, in some it is not (e.g. Fig6). Please. homogenise.

- Figure 5I-L should be moved to Figure 6

- In some figures, clones are sometimes indicated as “clon” (e.g. Fig7). Please correct.

**Have all data underlying the figures and results presented in the manuscript been provided?**

Reviewer #1: Yes

Reviewer #2: Yes

Reviewer #3: Yes

PLOS authors have the option to publish the peer review history of their article (what does this mean?). If published, this will include your full peer review and any attached files.

Reviewer #1: No

Reviewer #2: **Yes: **Michel Gho

Reviewer #3: No

---

## [Decision Letter · Decision Letter 1]

20 Sep 2023

Dear Dr Tejedor,

We are pleased to inform you that your manuscript entitled "A novel proneural function of Asense is integrated with the sequential actions of Delta-Notch, L’sc and Su(H) to promote the Neuroepithelial to Neuroblast transition" has been editorially accepted for publication in PLOS Genetics. Congratulations!

Yours sincerely,

Bassem A. Hassan, Ph.D.

Guest Editor

PLOS Genetics

Gregory P. Copenhaver

Editor-in-Chief

PLOS Genetics

Comments from the reviewers (if applicable):

Reviewer's Responses to Questions

**Comments to the Authors:**

Reviewer #1: All my comments has been reasonably addressed.

Reviewer #2: The authors have responded to all points I have raised and have incorporated these responses, where necessary, into the revised version of their manuscript.

My main issue concerned the role of Su(H) on the NE-NB transition and the possibility that the Notch signal acts independently of Su(H). The authors have made changes to narrow the scope of their conclusions in this revised version. They also enriched the figures as proposed in order to clarify their remarks. They presented convincing arguments to justify certain conclusions about the dynamics of the events studied, even when working with fixed material. Finally, they enriched the concluding figure to better explain how their model is consistent with a morphogenic wave that sweeps on the epithelium.

As such, I consider this revised version ready for publication, at least as far as my reviews are concerned.

Reviewer #3: The authors have sent a revised version of their manuscript describing a new Ase+ step in the transition from NE to NB in the Drosophila medulla. The new manuscript is clearer and contains additional data supporting the role of Ase as an important pro-neural factor in this context and consolidating the regulatory interactions with the Notch pathway.

I recommend publication.

**Have all data underlying the figures and results presented in the manuscript been provided?**

Reviewer #1: Yes

Reviewer #2: Yes

Reviewer #3: Yes

PLOS authors have the option to publish the peer review history of their article (what does this mean?). If published, this will include your full peer review and any attached files.

Reviewer #1: No

Reviewer #2: **Yes: **Michel Gho

Reviewer #3: No

**Data Deposition**

http://datadryad.org/submit?journalID=pgenetics&manu=PGENETICS-D-23-00385R1

**Press Queries**

---

## [Editor Report · Acceptance letter]

18 Oct 2023

PGENETICS-D-23-00385R1 

A novel proneural function of Asense is integrated with the sequential actions of Delta-Notch, L’sc and Su(H) to promote the Neuroepithelial to Neuroblast transition 

Dear Dr Tejedor, 

We are pleased to inform you that your manuscript entitled "A novel proneural function of Asense is integrated with the sequential actions of Delta-Notch, L’sc and Su(H) to promote the Neuroepithelial to Neuroblast transition" has been formally accepted for publication in PLOS Genetics! Your manuscript is now with our production department and you will be notified of the publication date in due course.

With kind regards,

Zsofi Zombor

PLOS Genetics

On behalf of:
